# Percolation transition of cooperative mutational effects in colorectal tumorigenesis

Dongkwan Shin[1], Jonghoon Lee[1], Jeong-Ryeol Gong[1] & Kwang-Hyun Cho[1]

Cancer is caused by the accumulation of multiple genetic mutations, but their cooperative effects are poorly understood. Using a genome-wide analysis of all the somatic mutations in colorectal cancer patients in a large-scale molecular interaction network, here we find that a giant cluster of mutation-propagating modules in the network undergoes a percolation transition, a sudden critical transition from scattered small modules to a large connected cluster, during colorectal tumorigenesis. Such a large cluster ultimately results in a giant percolated cluster, which is accompanied by phenotypic changes corresponding to cancer hallmarks. Moreover, we find that the most commonly observed sequence of driver mutations in colorectal cancer has been optimized to maximize the giant percolated cluster. Our network-level percolation study shows that the cooperative effect rather than any single dominance of multiple somatic mutations is crucial in colorectal tumorigenesis.

---

[1] Laboratory for Systems Biology and Bio-inspired Engineering, Department of Bio and Brain Engineering, Korea Advanced Institute of Science and Technology (KAIST), Daejeon 34141, Republic of Korea. Correspondence and requests for materials should be addressed to K.-H.C. (email: ckh@kaist.ac.kr)

Genes interact with each other epistatically, and multiple somatic mutations are often involved in tumorigenesis[1, 2]. However, little is known regarding their cooperative effect at the genome-wide level. Previous studies have shown that combinations of multiple mutations rather than a single mutation might play important roles in tumorigenesis. For instance, mutations of certain genes do not occur randomly but tend to co-occur in cancer patients[3, 4], and colorectal cancer develops through the sequential accumulation of driver mutations such as APC, KRAS, PI3K, and TP53[5–8]. This suggests that the cooperativity of driver mutations might be an important factor for the development and progression of cancer. There have been attempts to identify critical pairs of cancer-inducing genes by statistically examining their cooperative effect, but the hidden mechanism underlying such cooperative multiple mutations during tumorigenesis remains a perplexing puzzle.

Most genetic alterations in cancer influence signal transduction, which is under complicated regulation[8, 9]. Hence, the interaction of multiple somatic mutations should be investigated in the underlying molecular regulatory network. Co-occurring mutations in cancer are often found to be involved in different signaling pathways[3, 4, 10], so we can infer that they provide additive or even synergistic effects on the selective growth advantage of cancer. However, genes mutated in an exclusive way are likely to be involved in the same signaling pathway; these rarely confer any significant selective growth advantage on cancer due to the fact that the functional consequences of single mutations or double mutations are similar[3, 4, 10, 11]. These results suggest that whether a pair of somatic mutations is populated in the same signaling pathway or in distinct pathways might be crucial for tumorigenesis. This also highlights the need to investigate the cooperativity of somatic mutations upon the underlying molecular interaction network. Recent network-based, pan-cancer analysis studies showed that the topological location of somatic mutations in the protein–protein interaction (PPI) network might be closely associated with the clinical outcome. Hofree et al.[12] proposed such a network-based stratification of cancer patients based on The Cancer Genome Atlas (TCGA), where a network propagation of somatic mutations on the PPI network was introduced by assigning high values to non-mutated genes that are close to mutated genes. Another study proposed to identify significantly mutated sub-networks consisting of canonical signaling pathways and other parts that are rarely affected by mutations for each cancer type based on a heat diffusion model, wherein hot genes (frequently mutated genes such as TP53) propagate their heat to their neighboring genes[13]. These studies, however, provide only limited information regarding the influence of multiple somatic mutations on tumorigenesis, which involves the dynamical change of cooperative mutation effects upon the underlying molecular interaction network.

Percolation, a popular cooperative phenomenon in physical systems, describes the dynamical properties of various complex networks, where the system undergoes a critical transition in size or functioning during its growing process, called a percolation transition[14–16]. Examples include spreading of computer viruses on computer networks[17], rumor spread or information diffusion in social networks[14, 18], and epidemic spreading of infectious diseases over a network of towns[19]. To explore such a cooperative phenomenon of somatic mutations in tumorigenesis, we employ a network propagation method to measure the spreading of the influence of the somatic mutations on the molecular interaction network and then examined the cooperative effect of the somatic mutations by mapping all of the mutations observed in colorectal cancer from TCGA to a large-scale molecular interaction network. Throughout this network-level systems biological study, we find that a subnetwork area representing the cooperative effect of multiple somatic mutations forms a giant cluster (GC), which is the largest connected subnetwork in which all genes have mutation influence scores beyond a certain threshold. This cluster undergoes such a percolation transition during tumorigenesis, resulting in a giant percolated cluster (GPC) accompanied by phenotypic changes corresponding to cancer hallmarks. Intriguingly, we further find that the most commonly observed sequence of driver mutations in colorectal cancer has been optimized to maximize the GPC. Our findings provide new insight into the cooperative mechanism of multiple somatic mutations in colorectal tumorigenesis.

## Results

**Network propagation of somatic mutations and a GC**. We began by applying the network propagation[20] to somatic mutations from TCGA colorectal cancer patients on a PPI network ($N = 10{,}968$) to spread the influence of each mutation over the network neighborhood (see the Methods section). The network propagation has been used to stratify cancer patients[12], whereby influence scores are assigned to every node such that the nodes closer to a gene harboring a somatic mutation have higher scores. Therefore, when a mutation occurs on a node, its nearest neighbors have higher values, whereas nodes far away from the mutated node have values near zero. Thus, mutation influence appears to propagate along the PPI network. Indeed, it is expected that a mutation could not successfully affect all the nodes but at most cover a few layers of nearest neighbors. Thus, we can predict an effective boundary of mutation influences centered on the mutated node, inside which nodes have influence scores beyond a certain threshold and form a sub-network that we call a "mutation-propagating module" (Fig. 1a). In our study, we mapped expression profiles of each patient to the PPI network to determine the weight of links, such that patient-specific PPI networks were generated before projecting mutation profiles of patients to the network (see the Methods section). When mutation profiles of a patient are mapped to the patient-specific network, mutation-propagating modules of individual mutations are scattered throughout the network, often forming connected modules, which are sub-networks influenced by two or more mutations. Among them, the largest connected module, which we call "giant cluster (GC)", could largely cover the network in the order of network size, N, and, therefore, could have a significant impact on the network (Fig. 1a).

On the basis of this network propagation method, we first investigated the GC size for each patient by projecting mutation profiles to the respective PPI network. On average, the GC covered a relatively large portion (20–40%) of the entire network, although patients only have an average of 20–40 somatic mutations. The threshold of mutation influences was selected to be high, so that each mutation-propagating module approximately included only nearest neighbors (Fig. 1b). These indicate that multiple somatic mutations interact cooperatively to enlarge the effective boundary of mutation influences. To test whether such cooperative effects in GCs come from nonrandom mutation profiles, for each patient we compared the size $S_{gc}$ of its GC with the expected size $S_{gc}^{rand}$ if the same number of mutations occurred randomly on the network. Figure 1c illustrates that the GC size for patients was significantly larger than the random expectation, and such differences were prominent when the cancer-related genes[21] or driver genes[8] were considered rather than all the mutations (Supplementary Fig. 1). We also confirmed that the statistical significance of this tendency continues in many cases of different thresholds (Supplementary Fig. 2). Hence, these results suggest that the cooperation among somatic mutations in cancer cannot be attributed to a random selection of mutations but can

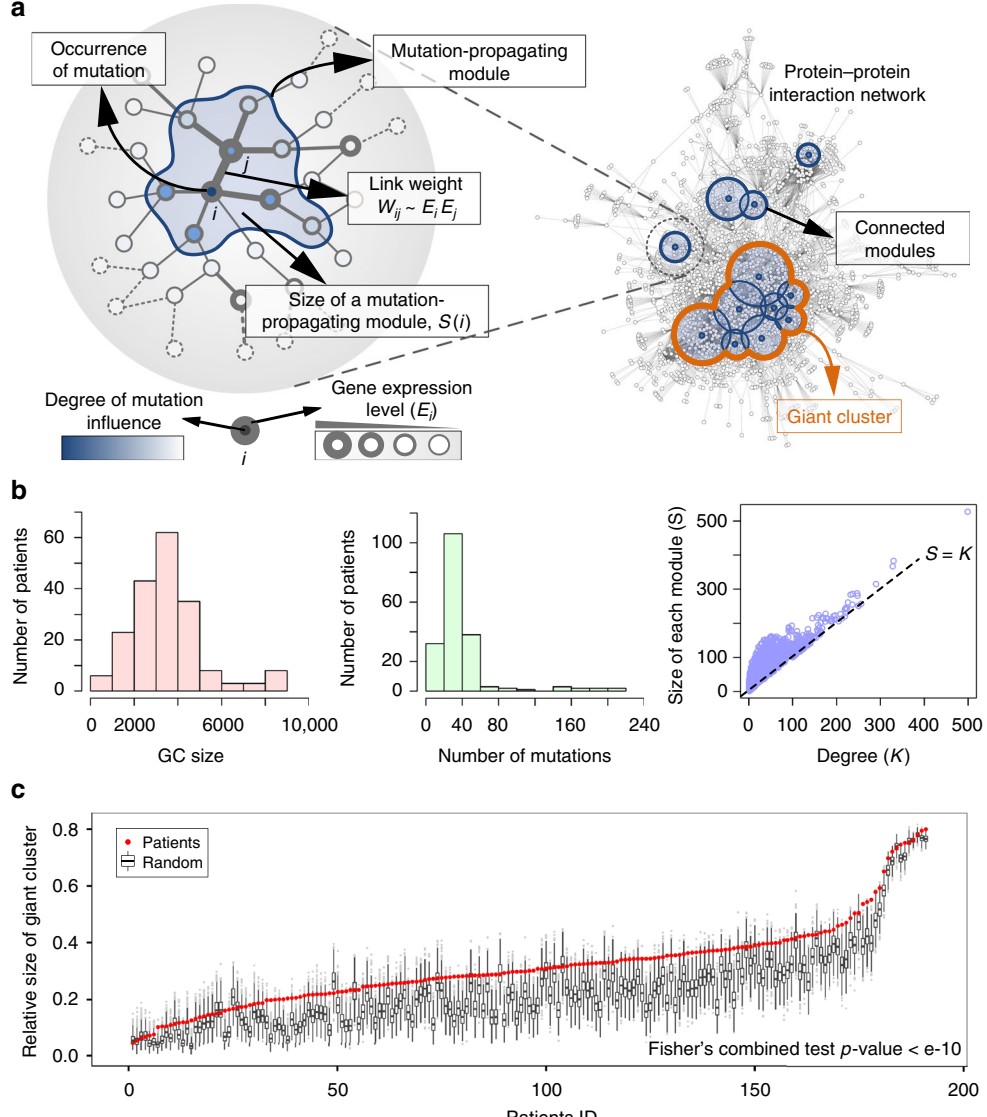

**Fig. 1** The cooperative mutation effects represented by a giant cluster upon a PPI network. **a** Formation of a giant cluster upon a PPI network by the propagation of mutation influences. A mutation influence propagates along a PPI network and forms a mutation-propagating module, which is a subnetwork that is effectively influenced by the mutation. The link weight is determined by the product of the expression levels of its end genes, $W_{ij} \sim E_i E_j$. Several mutation-propagating modules occasionally form connected modules or a giant cluster, which is the largest connected module among them. The color intensity of nodes represents the degree of mutation influence, and the width of a circle indicates the expression level of the corresponding gene. **b** Distribution of the size of GC (left) and the number of mutations (center) for 191 cancer patients. Right figure shows the size of each mutation-propagating module for all the mutated genes found in patients vs. the degree of each corresponding node for a threshold $V = 0.001$. **c** The size of GC normalized the network size for 191 patients compared to the random expectation ($n = 1000$) where the same number of mutations for each patient was randomly selected

be determined by topological properties of somatic mutations in the PPI network.

We further investigated the formation of a GC for other type of colon cancer (DFCI data set obtained from cBioPortal, $n = 526$)[22, 23] as well as eight other types of solid tumors (BLCA, BRCA, HNSC, KIRC, LUAD, LUSC, PAAD, and STAD obtained from TCGA). We compared the size of the GC between the colorectal cancer patient and the random expectation for which the same number of virtually mutated nodes as in each patient were randomly generated in a repeated way ($n = 1000$), and the averaged GC size was measured for each patient. Performing this comparison clearly shows that colorectal cancer can be characterized as having a significantly larger size of GC

(Supplementary Fig. 3, and see Methods). We also confirmed that the statistical significance of this tendency was also observed in the other types of cancer (Supplementary Fig. 3). As a result, we validated our hypothesis of the formation of a GC in various types of cancers.

**Overlap and synergy between mutation-propagating modules.**
We also asked if there is a relationship between the GC caused by somatic mutations from cancer patients and network characteristics of mutation pairs. We found that the average degree of genes that harbor somatic mutations in patients was significantly larger than that in random selection of nodes, whereas the average shortest distance between mutation pairs in patients was

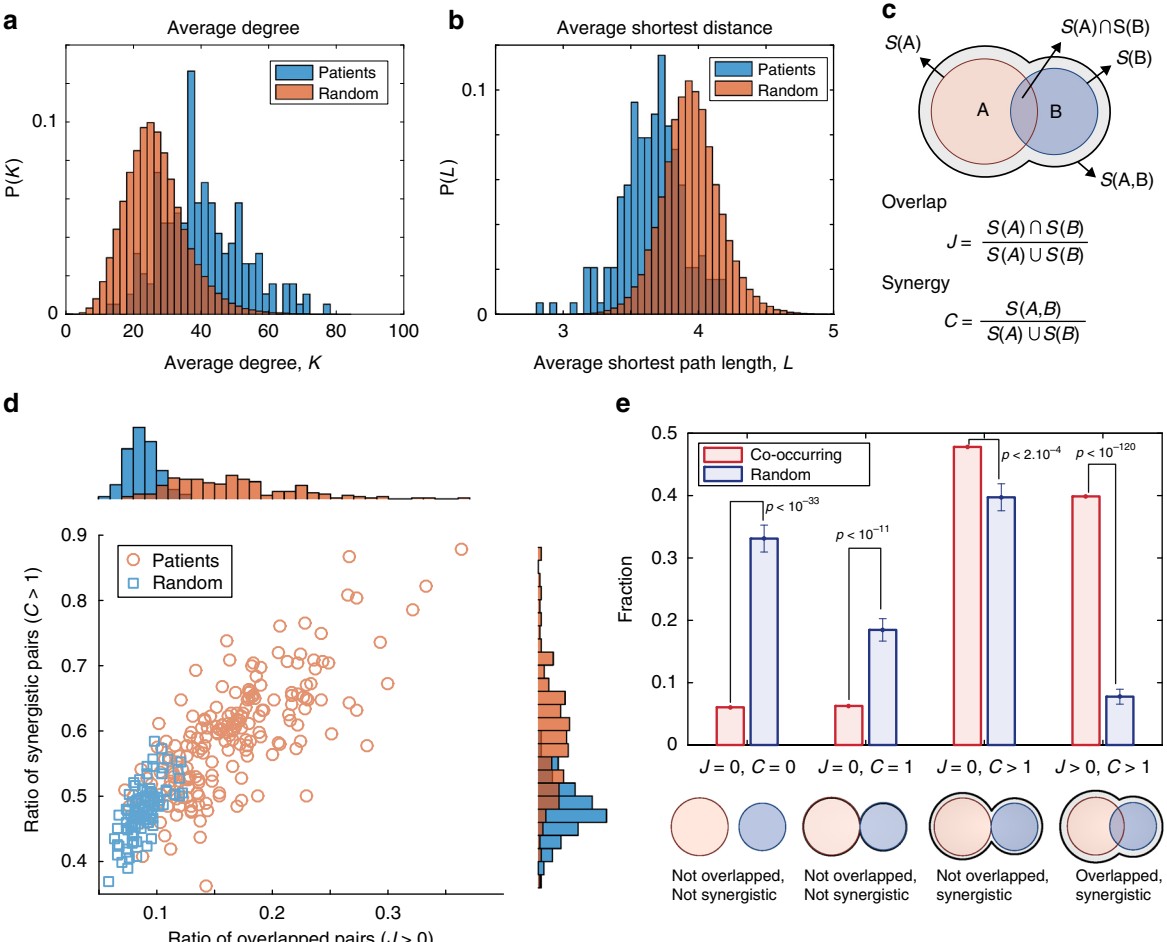

**Fig. 2** Overlap between mutation-propagating modules and their synergistic effects. **a** Distribution of the average degree of genes that harbor somatic mutations in patients compared to the corresponding random expectation. The average degree of each patient indicates the mean degree of all the mutated genes of that patient, and the random expectation indicates the expected mean when the same number of mutations occurred randomly ($n = 1000$). **b** Distribution of the average shortest path length between all the mutations in patients compared to the corresponding random expectation ($n = 1000$). **c** The overlap between two mutation-propagating modules, A and B, is measured by the Jaccard index $J = S(A) \cap S(B)/S(A) \cup S(B)$. Their synergy is measured by $C = S(A,B)/S(A) \cup S(B)$. $S(A)$ denotes the size of the mutation-propagating module when a mutation A occurs, and $S(A,B)$ indicates the size of the connected module when both mutations A and B occur. If A and B are close enough, $S(A,B)$ would be larger than the union of $S(A)$ and $S(B)$. Therefore, there can be some extended area (gray), which indicates additional nodes that are in neither $S(A)$ nor $S(B)$. However, if A and B are far enough apart such that their modules do not overlap, $S(A,B)$ would be 0. The values of $J$ lie in the range [0, 1], with $J = 0$ for no common genes and $J = 1$ for identical gene sets between two modules. $C > 1$ indicates that a connected module is larger than the simple union of the single modules A and B, we call "synergistic". **d** Ratio of synergistic pairs ($C > 1$) vs. ratio of overlapped pairs ($J > 0$) among all the pairs of somatic mutations for individual patients. For reducing the computational complexity, we considered only a case for randomly selected 100 mutations with 100 iterations. **e** The fraction of four types of connected modules within 479 co-occurring mutation pairs (see the Methods section for details), compared to the random case with the randomly selected same number of mutation pairs ($n = 1000$)

considerably shorter than the random expectation (Fig. 2a, b and Supplementary Fig. 4). It is probable that more neighbors of a mutated node would contribute to forming a larger mutant-propagating module. Moreover, the closer these modules are, the larger a connected module will be, resulting in the larger GC, as in Fig. 1c. To test this possibility, we introduced the overlap and synergy between two mutation-propagating modules, A and B, which are measured by the Jaccard index $J = S(A) \cap S(B)/S(A) \cup S(B)$ and $C = S(A,B)/S(A) \cup S(B)$, respectively (Fig. 2c). The size of the GC would be maximized if and only if all the pairs of mutation-propagating modules overlapped ($J > 0$) and their connected module also exhibited beyond the full coverage ($C > 1$), i.e., synergistic. We investigated the ratio of overlapped ($J > 0$) or synergistic ($C > 1$) pairs in each patient and found that the distributions of both ratios for patients shifted toward higher ratios compared to the randomly selected mutation sets (Fig. 2d), indicating that the larger GC for patients could possibly be attributed to either overlapped or synergistic mutation pairs. We also found that, for most patients, more than half of the mutation-propagation module pairs (55–70%) were synergistic, whereas relatively few pairs (10–20%) overlapped. To test whether this tendency was also observed in co-occurring mutation pairs across samples, we identified 479 mutation pairs with statistically significant co-occurrence among 382 driver genes from Vogelstein[8] (see the Methods section for details). Figure 2e shows that the fraction of overlapped and/or synergistic co-occurring pairs was significantly larger than expected by chance, and nearly half of the co-occurring pairs were not overlapped but synergistic, which is consistent with the previous studies in which co-occurring mutated genes participate in different signaling

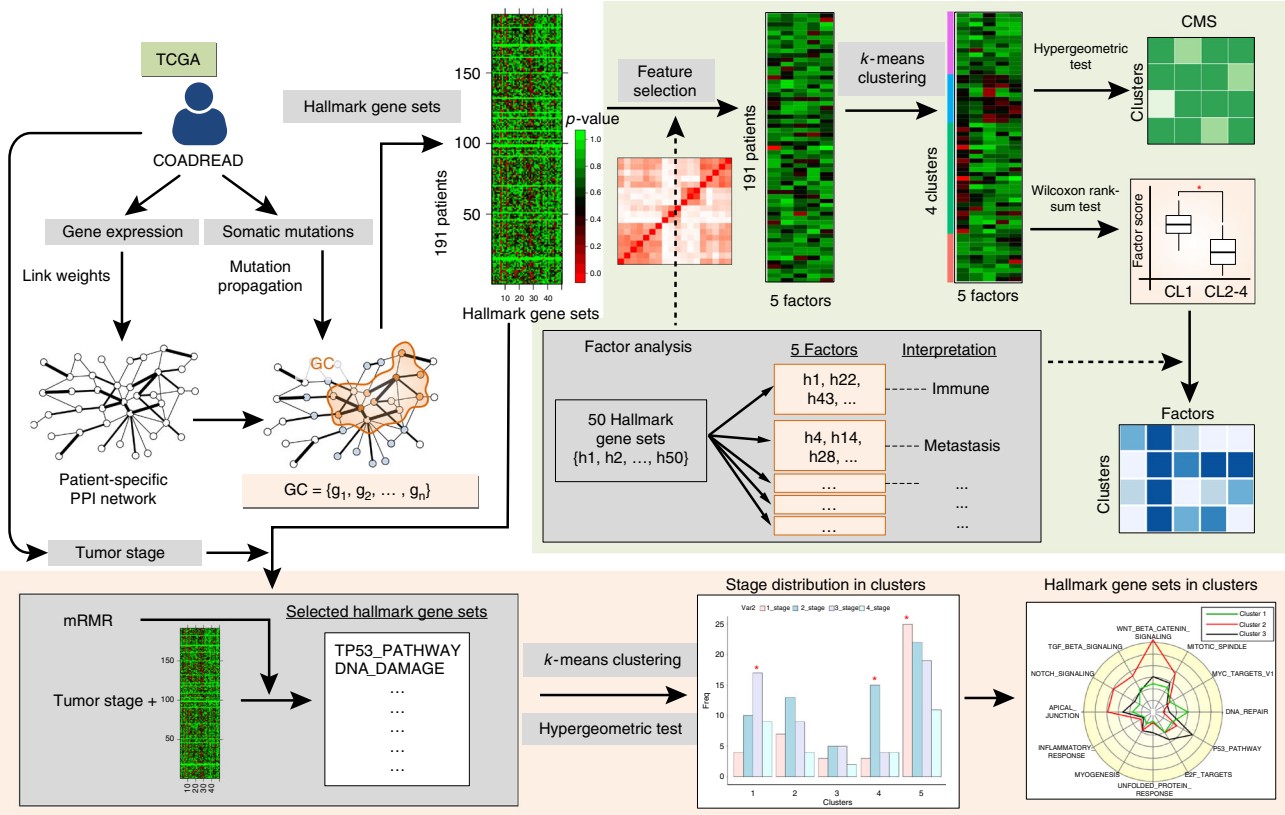

**Fig. 3** Hallmark gene set analysis. Flowchart of the hallmark gene set analysis for identifying hallmark gene sets enriched in the GC and for classifying cancer patients based on the selected features using factor analysis and the mRMR feature selection method (see Methods for details)

pathways[3, 4, 10]. Altogether, these results suggest that somatic mutations occur to minimize the overlap among mutation-propagating modules with the constraint under which the coverage of connected modules is maximized.

**Hallmark gene sets enriched in the GC.** If a GC comes from nonrandom mutation profiles of a cancer patient, genes within the GC must be correlated with any relevant biological and clinical characteristics of the cancer patient. To investigate which biological processes are inherent in the GC, we used a collection of 50 "hallmark" gene sets[24] (Supplementary Table 1) derived from the Molecular Signature Database (MSigDB) and identified hallmark gene sets enriched in the GC. We hypothesized that the cooperation of multiple somatic mutations in a GC can activate multiple cancer-related hallmarks that can cooperatively enhance tumorigenesis. For instance, the co-occurrence of a mutation that persistently activates a proliferative signaling and another mutation that induces invasion and metastasis will promote cancer malignancy. Therefore, it is important to find what kind of multiple hallmarks of cancer are enriched in a GC and more importantly, to find whether such phenotypic features have a real biological significance. In this respect, we attempted to interpret the phenotypic features contained in the GC with respect to the classification of tumors such as the consensus molecular subtypes (CMSs)[25] of colorectal cancer or tumor stages of cancer patients.

First, we examined how well the GC-based patient classification matches the previous CMS classification that is known to be the most robust molecular classification currently available for colorectal cancer[25] and further investigated, which key biological features the subtypes have (Fig. 3 and Methods). By propagating all of the mutations of each colorectal cancer patient, we obtained

a gene list included in the corresponding GC and estimated the enrichment of the hallmark gene sets in each patient through the hypergeometric test. To reduce the dimensionality of the resulting matrix of the hallmark gene set enrichment test (191 patients × 50 hallmark gene sets), we did factor analysis, which is often used in gene expression data for patient clustering as a robust feature selection method[26, 27] using standardized $z$-scores of $-\log(p\text{-value})$ for each hallmark gene set because the hallmark gene set variables have different scales. For an optimal predefined factor number ($k = 5$) (Supplementary Fig. 5a and see Methods for details), the correlation matrix of the hallmark gene sets were clustered into four global factors (Supplementary Fig. 5b, and see Supplementary Data 1 for various factor numbers, $k = 3, 4$, and 5) each of which can be characterized with several hallmark gene sets for a conventional weight threshold of 0.5: Factor 1 for angiogenesis and the metastasis pathway, Factor 3 for the immune response, Factor 4 for the Myc pathway and uncontrolled proliferation, and Factor 5 for the metabolic pathway (Fig. 4a). Interestingly, these factors not only contain important features related to cancer cells but also are very similar to the biological characteristics that distinguish each CMS group[25]. Factors 1, 3, 4, and 5 correspond to the characteristics of CMS4 (mesenchymal), CMS1 (MSI (microsatellite instability) immune), CMS2 (canonical), and CMS3 (metabolic), respectively. To investigate whether cancer patients can be classified into the CMS group by these factors, we performed statistical clustering ($k$-means) analysis on the factor scores of the patients. The result shows that the patient population can be clustered into four clusters, and interestingly, each cluster is strongly correlated with a distinct CMS group (Cluster 1—CMS2, Cluster 2—CMS4, Cluster 3—CMS1, and Cluster 4—CMS3) (Fig. 4b). For a biological understanding of the cluster groups, we then examined

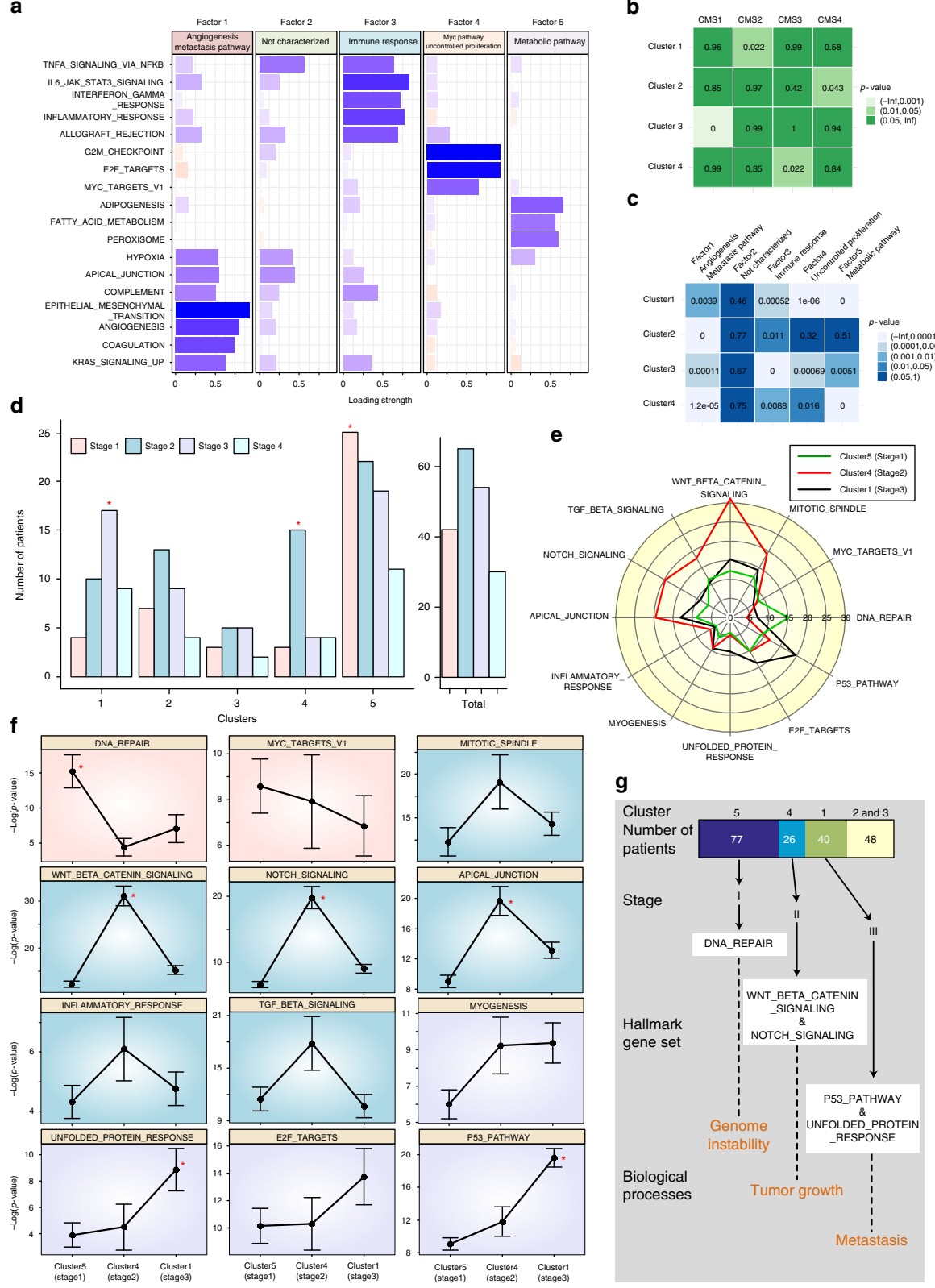

which Factors are dominant in each cluster (Fig. 4c) and compared the results with the biological signatures of the CMS groups[25]. Cluster 1 was mainly characterized as Factor 4 (Myc pathway and uncontrolled proliferation) and Factor 5 (metabolic pathway), while Factor 1 (angiogenesis and metastasis pathway) and Factor 3 (immune response) had a relatively low significance,

which is in good accordance with the biological characteristics of the CMS2 group except for Factor 5. Cluster 2 was mainly characterized as Factor 1 (angiogenesis and metabolic pathway) but not as Factor 4 (Myc pathway and uncontrolled proliferation), and Factor 5 (metabolic pathway). These features are very similar with the biological signatures found in CMS4, which

showed epithelial–mesenchymal transition-related signatures, but have less statistical significance with the signatures associated with Wnt and Myc targets, cell cycle, or metabolism. Cluster 3 was mainly characterized as Factor 3 (immune response) which is a key biological feature of the CMS1 group. Cluster 4 is mainly characterized as Factor 5 (metabolic pathway), which is in agreement with the enrichment of the metabolism signatures in CMS3. Taken together, our GC-based hallmark gene set analysis enables colorectal cancer patients to be clustered into informative subtypes for which biological features well match those of the previous CMS groups. Although extracted biological features of the GC depend on the threshold of mutation influences (Supplementary Figs. 6 and 7), we could confirm that the GC caused by somatic mutations of cancer patients is a biologically meaningful network index that represents a biological state or process during the development of cancer. Indeed, because most cancers have mutations in cancer-inducing genes, it is possible that cancer-related hallmark gene sets enriched in the GC come from such genes. However, when we applied the network propagation to only cancer-related or driver genes rather than all of the mutated genes of each patient and then executed the hallmark gene set analysis with varying thresholds, most of the hallmark gene sets that are associated with cancer progression or the CMS groups were not enriched in patients (Supplementary Figs. 8–13). These results suggest that the cooperative effect of multiple somatic mutations, including driver as well as passenger mutations, is critical in colorectal cancer progression.

Second, to investigate whether the hallmark gene sets enriched in cancer patients correlate with clinical tumor stages, we performed statistical clustering analysis of the enrichment score of the hallmark gene sets for cancer patients according to the significant hallmark gene sets. Twelve significant hallmark gene sets were selected to distinguish the tumor stages of cancer patients using the Minimum Redundancy Maximum Relevance (mRMR) feature selection[28, 29], an algorithm often used to identify relevant features that are mutually far apart while having a high correlation with the classification variable, for example, the tumor stage in this study (see Methods for details). The result revealed that the patient population can be divided into five clusters, and three clusters are strongly correlated with different tumor stages (Cluster 1—stage 3, Cluster 4—stage 2, and Cluster 5—stage 1), although 48 patients in Clusters 2 and 3 do not show any correlation with the tumor stage (Fig. 4d). Interestingly, the enrichment of a few hallmark gene sets is significant in each cluster associated with tumor stage, and there are biologically and clinically meaningful relationships between the tumor stages and hallmark gene sets: DNA_REPAIR in stage 1, WNT_BETA_CA-TENIN_SIGNALING, NOTCH_SIGNALING, and APICAL_-JUNCTION in stage 2, and UNFOLDED_PROTEIN_RESPONSE and P53_PATHWAY in stage 3 (Fig. 4e, f), indicate that there exist tumor stage-specific hallmark gene sets that are enriched in the GCs of cancer patients. In the early stage of cancer, cancer cells are often initiated by genomic instability due to the

dysfunction of DNA repair proteins, and become larger through hyperproliferation. Thus, they start to spread into nearby tissues or lymph nodes when entering the late stages. Our results also show such phenotypic changes according to tumor progression (Fig. 4g). A patient cluster characterized as stage 1 tends to show enrichment of the gene set related to DNA repair, which is known to cause genomic instability in the early stage of colon cancer[30, 31]. A patient cluster characterized as stage 2 shows enrichment of the gene set related to the Wnt/β-catenin and Notch signaling pathways, both of which cooperatively control cell proliferation and tumorigenesis in the intestine[32]. A patient cluster character-ized as stage 3 shows enrichment of the gene set related to the unfolded protein response, which induces metastasis through hypoxic activation[33, 34], and p53 signaling which influences metastasis by the dysfunction of p53[35]. These results also confirm our hypothesis that the formation of a GC implies the co-activation of multiple hallmark gene sets that are crucial for tumorigenesis. Taken together, we conclude that cancer patients could be classified into several subtypes according to a few hallmark gene sets involved in the tumor stages, consequently suggesting that the GC conveys biologically and clinically relevant phenotypes.

We further investigated whether the immune score based on the hallmark gene set analysis is well distinguishable between MSI and MSS (microsatellite stable) patients. For this, we introduced three immune scores that are defined with the statistical significance of specific hallmark gene sets associated with immune response, such as the immune process category from MSigDB, the immune response factor from our factor analysis, and carefully selected hallmark gene sets that are considered essential for the immune response. The immune scores that we defined show a statistically significant difference between the MSI and MSS patients (Supplementary Fig. 14 and see Methods), compared to the groupings based on several estimates related to immune response or tumor purity. Therefore, our results indicate that the immune scoring system based on the hallmark gene set can discriminate MSI and MSS colon cancer patients, confirming that the phenotypic features inherent in the GC have a real biological significance.

**Percolation transition of a GPC**. The giant component of a random network in percolation theory refers to a largely con-nected cluster that undergoes the sudden transition from small and disconnected clusters by the gradual addition of links[36]. The existence of such a large-scale connectivity is very important in real-world networks. Largely connected clusters in the global airline network or traffic network are essential for efficient transportation, whereas a well-connected social or computer network can be badly utilized for virus spreading. Emergence of such a percolating cluster in physical networks often indicates that the system goes through a percolation transition, where significant changes may occur in the physical properties of the

**Fig. 4** Patient classification and biological interpretation of the patient clusters. **a** The correlation matrix of the hallmark gene sets was clustered into four global factors with biological characteristics. Each bar indicates the loading strength of a hallmark gene set in each factor. Blue (red) bars represent positive (negative) values, and absolute values were used for the negative values. **b** The four clusters were classified with statistical clustering (k-means) analysis of the factor scores of the patients. The hypergeometric test was performed to examine the statistical significance of the enrichment of individual CMS groups in each cluster. **c** Biological interpretation of the clusters with the identified Factors, each of which corresponds to a biological signature of a CMS group. The distribution of the Factor scores of the patients in one cluster was compared to that of the other remaining clusters. p-values were obtained by performing the Wilcoxon rank-sum test. **d** Distribution of the tumor stages in each cluster. Red asterisks indicate statistical significance (hypergeometric test, $p < 0.05$). **e** Comparison of average values of $-\log$ (p-value) of each cluster in individual significant hallmark gene sets. **f** Comparison of the statistical significance ($-\log$ (p-value)) of three clusters in each significant hallmark gene set. Red asterisks indicate that there are significant differences in the statistical test results between a group with the highest value and the other two groups (Wilcoxon rank-sum test, $p < 0.05$). Error bars indicate the standard error. **g** Summary of the relationships between the hallmark gene sets and the tumor stages in the individual cluster

system. For instance, in a lattice network that consists of conducting and non-conducting subunits between metallic plates, the electrical current can flow through the percolating conducting subunits above, but not below, the percolation threshold[37]. In our work, scattered, mutation-propagating modules formed a GC in a PPI network; more importantly, genes within the GC represented the phenotypic properties of cancer, including cell proliferation and metastasis. Therefore, in the context of percolation phenomenon, we propose that the GC resulting from somatic mutations of a cancer patient could be referred to as a "giant percolated cluster" (GPC), which is the largest percolating cluster that integrates the influences of scattered somatic mutations so that it confers phenotypic changes corresponding to cancer hallmarks. Then, one fundamental question naturally arises: whether the GPC, in reality, would undergo a percolation transition on the accumulation of somatic mutations during cancer progression.

To address this question, we hypothesized that the dynamic behavior of the GPC in tumorigenesis would depend on detailed selection rules of somatic mutations, where the accumulation of somatic mutations, corresponding to the gradual addition of links in the percolation study of a random network, can be considered as a discrete time axis. We introduced three different selection rules to determine which somatic mutation would be added during cancer progression. We then simulated changes in the size of the GPC by adding a somatic mutation selected from among the mutation profiles of cancer patients according to the rules at each evolution step (Supplementary Fig. 15 and see Methods for further details). The first selection rule was designed to minimize the overlap between mutated genes; in this way, cancer progression could be suppressed by preventing the cooperation between somatic mutations (Fig. 5a). Therefore, a somatic mutation added at the next time step minimized the degree of overlap with previous mutations, where the degree of overlap between a pair of somatic mutations was defined as high when they had high node degrees and were interconnected by short network distances (Supplementary Fig. 16). The second selection rule was to select the next mutated gene such that the size of the connected module between a pair of mutation-propagating modules was maximized (Fig. 5b), which represents the mechanism facilitating cancer progression by maximizing the cooperative effect of somatic mutations. In contrast, the last selection rule was to select the next mutated gene, minimizing the size of connected modules, among mutation candidates in which their mutation-propagating modules overlapped with those of previously mutated genes, $J > 0$ (Fig. 5c). This rule describes two opposite or contrary driving forces in the development of cancer. One is regarding the suppressive process that prevents cancer progression by minimizing the cooperation of somatic mutations, and the other is regarding the process that promotes cancer progression by ensuring connections with any previously mutated genes.

On the basis of these rules, we identified all available mutation sequences from mutation profiles of colorectal cancer patients and investigated the order of pairs of key driver mutations frequently found in colorectal cancer, including APC, TP53, KRAS, PIK3CA, and SMAD4[5–8]. Interestingly, the order of driver mutations tended to follow the well-known sequence of driver mutations when the first rule, which suppresses the overlap between a pair of mutations, was applied (Fig. 5d). APC mutations occurred ahead of other driver mutations; TP53 mutations always occurred as the last event among driver mutations; KRAS mutations was earlier events than TP53, PIK3CA, and SMAD4; and both PIK3CA and SMAD4 mutations occurred earlier than TP53, with high probabilities. In contrast, the second rule designed to maximize the size of connected

modules was found to mostly induce the opposite order (Fig. 5e). We also found that the last rule, which is involved in both cancer promoting and suppressive processes, determined a similar order of driver mutations as the first rule (Fig. 5f). Given that most early-stage colonic adenomas exhibited genetic alterations in APC[7], and most colorectal cancer patients with only a single driver mutation had APC mutations (Supplementary Fig. 17), these relationships between early and late mutation events can be reorganized into the complete mutation sequences available from APC to TP53 (Fig. 5g), confirming the most commonly observed sequences of driver mutations, e.g., $APC \rightarrow KRAS \rightarrow PIK3CA \rightarrow TP53$ and $APC \rightarrow KRAS \rightarrow SMAD4 \rightarrow TP53$[6, 38]. With these various sequences of somatic mutations determined by different selection rules, we further investigated the changes in GPC size along with the accumulation of somatic mutations in each patient. Intriguingly, we found a surprising result that the GPC size under the second rule increased at a relatively early stage, whereas the onset of the GPC could be delayed in the first and last rules, and the mutation sequence frequently found in colorectal cancer, e.g., $APC \rightarrow KRAS \rightarrow SMAD4 \rightarrow TP53$ in this patient, induced a dramatic increase in the GPC size (Fig. 5h). These results indicate that, although the most commonly observed sequence of mutations in colorectal cancer is determined by the mechanism that suppresses the cooperative effect among mutations, such a sequence finally accelerates the increase in the size of the GPC to maximize the cooperative effect, therefore inducing the sudden transition during the development of cancer. Furthermore, we also found that the system can undergo a percolation transition in which the size of the GPC under the last rule was suddenly increased compared to random selection, showing gradual increases of the GC (Fig. 5h). Taken together, our results have substantial implications for understanding the evolutionary process of tumorigenesis: (i) the GPC caused by cooperative effects of somatic mutations could undergo a percolation transition in colorectal tumorigenesis through the interplay between cancer suppressive mechanisms that minimize their cooperation and cancer promoting mechanisms that increase the size of the GPC, and (ii) the most commonly observed sequence of driver mutations in colorectal cancer could be essential for the sudden transition during tumorigenesis, consequently leading to the maximization of GPC size. This tendency was also observed in most patients who had few driver mutations (Supplementary Figs. 18 and 19), suggesting that the cooperation of passenger mutations as well as driver mutations may play an important role in inducing the percolation transition, even in colorectal cancer patients with few driver mutations.

## Discussion

Tumorigenesis is an evolutionary process in which the transition of a normal cell to a cancer cell is mediated by the accumulation of somatic mutations[39, 40]. In this process, the transition might not be gradual; not all of the somatic mutations directly contribute to the development of cancer, which is usually initiated after sufficient accumulation of mutations. Moreover, previous studies have shown that half or even more of the somatic mutations observed in cancer of self-renewing tissues actually occur before tumor initiation[40], and sequential accumulation of driver mutations accompanied by APC is required for the onset of colon cancer in many cases[6, 38]. Altogether, these suggest that a critical transition might be an intrinsic feature of the cancer evolutionary process. In this study, we found that a GC, the cooperative mutation effects represented by a large connected component upon a PPI network, might undergo a percolation transition during tumorigenesis. In normal tissue development, a number of somatic mutations can occur at various places in the

# ARTICLE

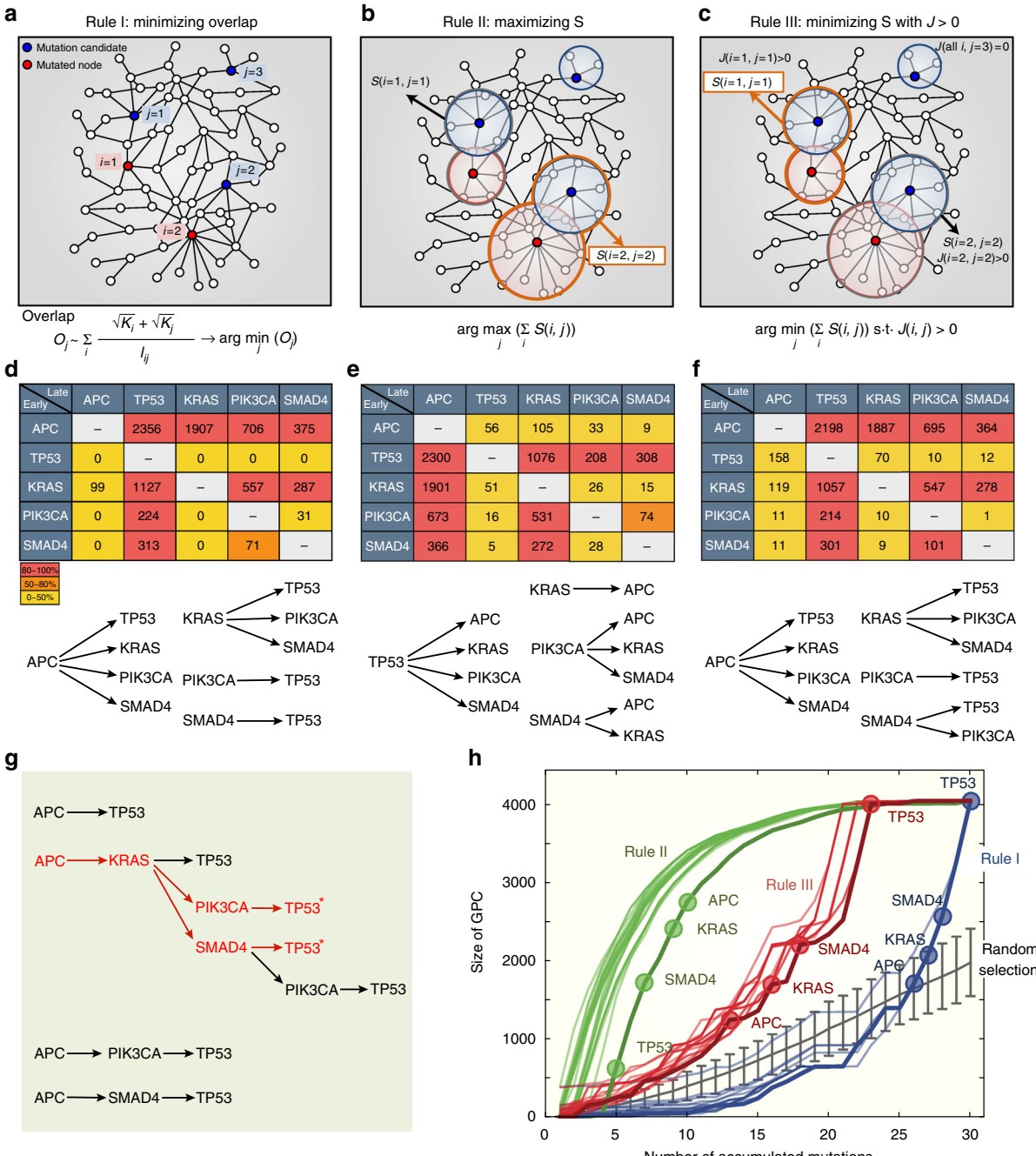

**Fig. 5** Percolation transition of a GPC. **a** A mutation selection rule for minimizing the degree of overlap between somatic mutations (see Methods and Supplementary Fig. 16 for details). The rule is to select a mutation candidate $j$ that minimizes the sum of overlap measures between $j$ and all the previously mutated ones. **b** A mutation selection rule for maximizing the size of connected modules. The rule is to select a mutation candidate $j$ that maximize the sum of $S(i, j)$ for all the previously mutated ones. For instance, node $j = 2$ will be selected as the next mutation because $S(i = 2, j = 2)$ is larger than $S(i = 1, j = 1)$ in the figure. **c** A mutation selection rule for minimizing the size of connected modules with the constraint that two modules of $j$ and a mutated one $i$ should overlap, $J(i, j) > 0$. For instance, node $j = 1$ will be selected as the next mutation because $S(i = 1, j = 1)$ is smaller than $S(i = 2, j = 2)$ in the figure. By applying the rules to the mutation profiles of individual patients, we obtained totally 3834 mutation sequences according to which mutation was selected as a seed. By investigating the order of a pair of driver mutations in the resulting mutation sequences, we constructed a matrix showing the number of mutation sequences that one driver mutation in a row occurs earlier than the other driver mutation in a column according to the first **d**, second **e**, and the last **f** rules. The bottom figures represent possible orders of driver mutation pairs with significant percentages (80–100%) in the respective rules. **g** All available mutation sequences from *APC* to *TP53* from the result of **f**. Red asterisks indicate the most commonly observed sequences of driver mutations in colorectal cancers. **h** The changes in the size of the GPC along with the accumulation of somatic mutations according to the rules, as an example, for a patient who has four driver mutations, *APC*, *KRAS*, *SMAD4*, and *TP53*, among 29 somatic mutations (see Methods for details). Driver mutations are denoted by circles at the corresponding order of occurrence of mutations in each rule. For comparison of the rules and the random expectation, we generated 100 mutation sequences among 29 randomly selected genes

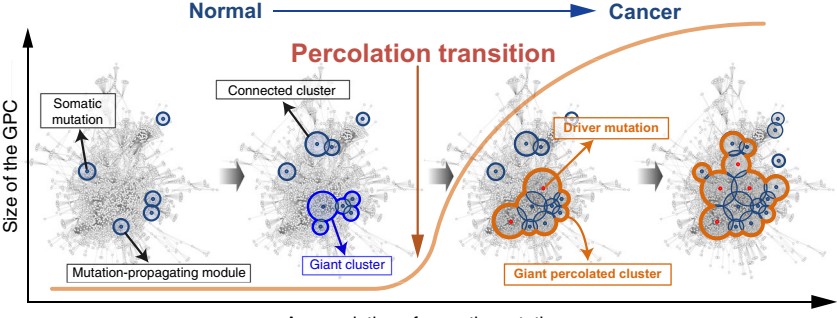

**Fig. 6** A schematic of a percolation transition of cooperative mutational effects during tumorigenesis

PPI network, and the resulting mutation-propagating modules can form a connected module or develop a GC along with the accumulation of somatic mutations. In this process, tumorigenesis can be initiated by a certain driver mutation that connects scattered clusters into one, leading to the formation of a GPC that represents cancer hallmarks (Fig. 6). Intriguingly, we found that the most frequently observed sequence of driver mutations characterizing colorectal cancer development might have been optimized to maximize the GPC. This finding provides novel insight into the relationship between cancer development and percolation transition, which can be useful for understanding the fundamental mechanism of tumorigenesis and further identifying new drug targets for anti-cancer therapy by interfering with GPC formation.

An interesting observation of our work is that the hallmark gene set analysis based on the GPC can extract all the features of the CMS groups, such as immune responses, proliferation, metabolism, and metastasis even though factor analysis was used, which is a type of unsupervised learning for feature selection or for data reduction. Moreover, the patient classification with these selected features shows very strong correlations with the CMS groups. There might be a few major reasons why our analysis based on somatic mutation profiles shows a consistent result with the CMS grouping derived from the gene expression profiles. First, we considered somatic mutations which are tumor-specific genetic changes, unlike germline mutations. Therefore, we were able to exclude some external factors that may be unrelated to cancer. Second, by considering the mutation influence on the PPI network, we were able to extract genes that might have influences on their expression levels by somatic mutations. A recent study suggested that mutation patterns correlate with global gene expression levels, and co-occurring driver mutation pairs tend to induce greater degrees of overlap in downstream transcriptional changes[41]. Our result shows that considering cancer-related or driver mutations alone could not capture all the features of the CMS groups (Supplementary Figs. 8–13), suggesting that the so-called passenger mutations will also have important roles in determining the functional activity of genes that are related to tumor progression. Third, the biased propagation of mutation influences based on a patient's gene expression profile better reflects the current state of the patient. Fourth, the introduction of an appropriate threshold to mutation influences enables us to determine an optimal gene set that can capture the inherent characteristics of cancer. Taken together, our results imply that a set of essential genes contained in the GPC obtained from the network propagation of somatic mutations can capture the important cancer-related features in the gene expression.

Considering the functional consequences of somatic mutations in cancer under the molecular network framework can be a promising strategy for not only understanding the cooperative

effects of multiple somatic mutations during tumorigenesis but also for identifying optimal targets for anti-cancer therapies. In a recent study by the Ideker Lab[12], have successfully stratified cancer patients based on a network propagation method. However, this network-based analysis differs from ours in many respects, especially in examining the dynamical change of cooperative mutation effects in tumorigenesis. First, we focused on an effective boundary of mutation influences around the mutated nodes and thus, could extract a sub-network for a mutation or a giant component for multiple mutations. Although the influence scores for a mutation can be assigned to all the nodes in the PPI network as in the study by Hofree et al., the mutation could not successfully affect the entire network but could at most cover a few layers of the nearest neighbors. Therefore, by considering such an effective boundary, we were able to obtain a GPC for the mutation profile of each patient and further confirmed the critical transition of the GPC during the development of cancer. Second, in addition to the mutation profile, the gene expression data of patients were also reflected in the PPI network, thereby obtaining the patient-specific network. Third, by spreading the influence of somatic mutations over the patient-specific network, we were able to extract key phenotypic characteristics that can explain the cancerous state of each patient and to observe phenotypic changes during the cancer development.

Our study showed that driver mutations are essential for triggering the onset of the GPC and also showed the significance of the cooperation of somatic mutations, including those not considered to directly affect tumorigenesis. All the biological features of the CMS groups were not enriched in GPC when only cancer-related or driver mutations were considered (Supplementary Figs. 8–13). Moreover, most cancer patients showed the percolation transition (or similar behavior), even in cases when they had only a few driver mutations (Supplementary Figs. 18 and 19), implying that the cooperative effect of driver and passenger mutations might play an important role in the development of cancer. Although many passenger mutations do not directly contribute to tumorigenesis, our results showed that the mutation effects of various passenger mutations found in individual cancer patients can converge to several cancer-related signaling pathways in which driver mutations occur, leading to the formation of a GPC and eventually contributing to tumorigenesis.

The accumulation of mutations in tumorigenesis causes a cancer to diverge into different clones, resulting in subclonal diversification of the tumor cells[42, 43]. Recent studies in breast cancer[42] and 12 different types of cancers from the pan-cancer analysis[44] showed that the mutation profiles can be very different among the subclones. Multiregion sequencing studies have also shown that differential mutation profiles exist across the subregions in some tumor samples, although some driver mutations are shared in different subclones[43, 45]. Therefore, we can predict that the formation of a GPC will be affected by the tumor

heterogeneity because the formation of a GPC is determined by the mutation profile of a cancer. To explore this, we further investigated whether the GPC size depends on the tumor heterogeneity by using the mutation profiles obtained from the multiregion biopsies of the primary regions from colorectal cancers with whole-exome sequencing[46]. The result suggested that as the tumor heterogeneity of a bulk tumor increases, the effect on the formation of the GPC becomes larger (Supplementary Fig. 20 and see Methods). If whole-exome sequencing of individual subclones is publicly available, we can analyze both the GPC for each subclone and the cancer hallmarks contained within it, which will provide further insight into understanding tumor heterogeneity and cancer evolution.

By illuminating the GPC, which represents the cooperative effects of the somatic mutations in cancer, our results suggest that identifying optimal targets to fragment the GPC into small pieces can provide a novel therapeutic strategy for the treatment of cancer. The fragmentation of GPC for the cancer treatment not only means breaking the network of the GPC itself, but it also means interfering with cancer-related phenotypes, such as cancer hallmarks, inherent in the GPC. For instance, if a cancer patient has a metastasis-related hallmark enriched in the GPC, we can select minimal targets that can break the GPC among the gene list related to that hallmark. Most cancers have multiple cancer hallmarks that can cooperatively promote tumorigenesis. Thus, we can consider some drug combinations, for example, one that interferes with the uncontrolled proliferation-related hallmark and the other that inhibits the metastasis-related hallmark. In recent studies on the percolation transition, there has been great interest in controlling the location of the percolation transition to delay its onset[47, 48]. Most mutation profiles of cancer patients available from databases, such as TCGA, only convey information regarding the final state, long after the onset of a GPC through cancer progression. Therefore, mutation profiles from the cancer database are not sufficient to develop a fundamental therapeutic strategy that can defer the onset of a GPC. Hence, the development of a mathematical model[49] that describes the evolutionary process of tumorigenesis along with the accumulation of somatic mutations may help reveal how to control the complex molecular regulatory network by identifying minimal and optimal intervention points to delay the onset of a GPC[50]. This can ultimately offer further insights into the development of human cancer and provide new opportunities for developing molecular therapeutics of a different concept.

## Methods

**Description of a PPI network and patient mutation profiles**. Somatic mutation data for colorectal cancer patients were obtained from the Firehose website (http://gdac.broadinstitute.org/runs/analyses__2014_04_16/reports/cancer/COADREAD-TP/MutSigNozzleReportCV/COADREAD-TP.final_analysis_set.maf). We first converted the genomic coordinates of the mutations to the hg19 assembly of the Human Reference Genome with the Liftover program (http://genome.ucsc.edu/cgi-bin/hgLiftOver) and then re-annotated with the Ensembl database using Annovar[51]. To retain only high-confidence pathogenic variants, mutations were filtered based on their predicted pathogenicity derived from five functional mutation prediction tools using Annovar. For nonsynonymous SNV, we included mutations that fulfilled at least two of the following five conditions: (i) SIFT[52] prediction class with "deleterious"; (ii) Polyphen2 HVAR[53] with "probably damaging" or "possibly damaging"; (iii) MutationTaster[54] with "disease causing automatic" or "disease causing"; (iv) MutatonAssessor[55] with "high" or "medium"; (v) CADD[56] Phred-score with 20 (top 1% of predicted damaging effect). For other mutation classes, we only considered stopgain, frameshift deletion, frameshift insertion, and frameshift substitution as pathogenic. To construct patient mutation profiles, we assigned the filtered genomic mutations to tumor samples by abstracting binary event calls such that a genomic event either occurred ("1") or did not occur ("0") in a gene for a given sample. We selected patients having less than 300 mutations from the 223 TCGA colorectal cancer patients to discard outliers having a very high number of mutations that result in a GPC covering the entire network, which left 198 patients. The expression data for colorectal cancer patients were downloaded from the Firehose website (http://gdac.broadinstitute.org/runs/stddata__2014_04_16/data/

COADREAD/20140416/gdac.broadinstitute.org_COADREAD. Merge_rnaseqv2__illuminaga_rnaseqv2__unc_edu__Level_3__RSEM_genes_normalized__data.Level_3.2014041600.0.0.tar.gz). Among them, 191 cancer patients for whom both mutation and expression profile information were available were selected. Patient information that includes the tumor stages, CMS, and MSI/MSS information of the data set we used was provided (Supplementary Data 2). The PPI network (N = 12,233) was previously constructed from STRING v9.0 by Hofree et al.[12]. We considered only the largest connected subnetwork (N = 12,071), due to the fact that network propagation is not available between a pair of nodes that does not connect with each other either directly or indirectly. By integrating the expression and mutation profiles of 191 cancer patients with the PPI network, we finally obtained the adjacency matrix of the PPI network (N = 10,968) and a patient-by-gene matrix that displays the mutation profiles of binary (1, 0) states on 10,968 genes for 191 patients. Somatic mutation data and RNA-seq gene expression data for breast invasive carcinoma (BRCA), bladder urothelial carcinoma (BLCA), head and neck squamous cell carcinoma (HNSC), kidney renal clear cell carcinoma (KIRC), lung adenocarcinoma (LUAD), lung squamous cell carcinoma (LUSC), pancreatic adenocarcinoma (PAAD), and stomach adenocarcinoma (STAD) were downloaded from the Firehose data run (https://confluence.broadinstitute.org/display/GDAC/Dashboard-Stddata). For our analysis, we only considered samples that had both RNA-seq and somatic mutation data. The somatic mutation data from the Dana-Farber Cancer Institute (DFCI) were downloaded from the cBio-Portal website[22, 23]. To retain only high-confidence pathogenic variants, mutations were filtered based on their predicted pathogenicity derived from five functional mutation prediction tools using ANNOVAR. The DFCI data set was used, which includes the somatic mutation profiles for a large population of patients but lacks the gene expression information. To compensate for the lack of gene expression information, the average expression profile of the TCGA colorectal cancer patients was used to extract the colon cancer specific average PPI network.

**Requirements for data sets**. In our study, we first mapped the gene expression profiles of individual patients to a large-scale PPI network to obtain patient-specific PPI networks and then, projected the somatic mutation profile of an individual patient to each network to explore phenotypic features embedded in the GPC. For this approach, there are three requirements for data sets. First, the data set should include both gene expression and somatic mutation profiles of colorectal cancer patients. Second, massive somatic mutations identified by whole-exome sequencing are required for the GPC analysis. One of the important implications of our findings is that although driver mutations are essential for triggering the onset of the GPC, passenger mutations also contribute to the formation of the GPC and eventually to the development of cancer. Therefore, a mutation data set profiled by whole-exome sequencing is more suitable for this study than a data set obtained by targeted-exome sequencing, which is often used for identifying specific mutations related to cancer. Third, the data set should include clinical outcomes such as tumor stages. Thus, based on these requirements, we chose from among the many genomic data sets TCGA data not only because it is the largest publicly available database for cancer-genomic studies, but also because it provides both messenger RNA and whole-exome sequencing data in addition to clinical information on the cancer patients. As a result, from the TCGA colorectal cancer data set, we obtained gene expression data for 263 patients and somatic mutation data for 223 patients. By examining the distribution of the number of mutations, we found that there are a few patients with a very large number of mutations (Supplementary Fig. 21). Considering that the GPC size of those patients having about 200 mutations already reaches about 80% of the entire network (Supplementary Fig. 2), it is evident that the GPC of patients having more than 300 mutations will cover the entire network. Hence, such cases will make it difficult to extract any statistically significant hallmark gene set. For this reason, patients with more than 300 mutations were excluded, resulting in 198 patients. Among those, 191 cancer patients for which both mutation and expression profile information are available were finally selected.

**The lists of cancer-related genes**. A typical tumor contains several cancer genes that can promote tumorigenesis. To investigate the functional consequences of the network propagation on cancer gene mutations, we extracted the cancer gene list in each cancer patient by using two collections of cancer-related genes. One is a list of 2102 cancer genes collected by Bushman Lab[21] (http://bushmanlab.org/links/genelists), which was obtained as a union of eight different data sets. Among them, we used 1687 cancer genes that intersected with the PPI network. Cancer genes comprised approximately 20–25% of all mutated genes in each patient. The other collection was a 418 cancer driver gene list, which was reported in a publication by Vogelstein and collaborators[8]. Among them, we used 382 cancer driver genes that intersected with the PPI network and that constitute approximately 5–10% of all the mutated genes in each patient.

**Network propagation of mutation effects**. To simulate the propagation of mutation effects through the PPI network, we employed the network propagation method[20], which describes a random walk with restart on a network. When a mutation occurs on a gene in a PPI network, a value "1" is initially assigned to the mutated gene. It then propagates along the network neighborhood such that higher

values are assigned to non-mutated genes that are closer to the mutated gene, according to the following equation:

$$F_{t+1} = \alpha A' F_t + (1-\alpha)F_0,$$

where $F_0$ denotes a matrix of binary (1, 0) states on genes, in which a "1" or "0" indicates whether a gene is mutated or not in the corresponding patient, $A'$ denotes a degree-normalized adjacency matrix of the PPI network, and $\alpha$ determines the degree of diffusion of a mutation influence throughout the network. We used an optimal value ($\alpha = 0.7$) for the network constructed from STRING v9.0, which were also used in a previous study by Hofree et al.[12]. In our study, changing $\alpha$ has a similar effect to changing the threshold $V$ of the mutation influences with respect to the formation of a GC. As $\alpha$ increases for a fixed value of $V$ (or $V$ decreases for a fixed value of $\alpha$), the size of the resulting GC increases. We considered the cases of various thresholds with a fixed value of $\alpha$ in the analysis of both the GPC and the hallmark gene set (Supplementary Figs. 2, 6, and 7), and confirmed that the main results do not change significantly. Here, the link weight of $A'$ is normalized by the degrees of its end-points. Therefore, we set $A' = D^{-1/2}AD^{-1/2}$ where a diagonal matrix $D$ is defined such that $D(i, i)$ is the sum of row $i$ of an adjacency matrix $A$ of the PPI network. An element of the adjacency matrix, $A_{ij}$, represents the probability of an interaction between node $i$ and $j$, along which the mutation influence propagates. Although two cancer patients have the same mutation profile, their resulting expression profiles must be completely different from each other. Therefore, we assumed that the probability of interaction between gene $i$ and $j$ in a given sample is proportional to the product of expression values of both genes in that sample and used an alternative to the adjacency matrix as $A_{ij} \rightarrow A_{ij}E_iE_j$, where $E_i$ indicates the expression value of gene $i$. By integrating expression profiles of each patient with the PPI network, we obtained a patient-specific PPI network, consequently enabling the implementation of more realistic propagation of somatic mutations on the patient-specific PPI network.

**Identification of co-occurring mutation pairs.** To identify co-occurring mutation pairs among 382 cancer driver mutations, we calculated the odds ratio for each pair of driver mutations and estimated the statistical significance using the Fisher's exact test. The odds ratio is given by the simple equation $OR = (A \cdot D)/(B \cdot C)$, where $A =$ number of patients altered in both genes, $B =$ number of patients altered in gene 1 but not gene 2, $C =$ number of patients altered in gene 2 but not gene 1, and $D =$ number of patients altered in neither gene[22]. Two-sided Fisher's exact test was used to produce $p$-values, and only $p < 0.05$ was considered significant. $OR > 1$ indicates co-occurring mutation pairs, whereas $OR < 1$ implies mutually exclusive mutation pairs. We identified 479 co-occurring and 14 mutually exclusive mutation pairs with statistical significance.

**Identification of biological features by the factor analysis.** To explore the biological functions that are enriched in the GC of each patient, we examined the enrichment of hallmark gene sets by hypergeometric test. For a gene list included in the GC, let $h$ be the number of genes annotated to a certain hallmark gene set, and let $N$ and $g$ be the network size and the number of genes in the GC, respectively. Suppose that the GC has $x$ genes annotated to this hallmark gene set, we can model $x$ by a hypergeometric distribution under the null hypothesis that a gene annotated to the hallmark gene set and a gene in the GC are independent events. Then, the $p$-value that measures the significance of enrichment is the probability of observing $x$ or more genes annotated to the hallmark gene set in the GC,

$$p-\text{value} = \sum_{k=x}^{\min(h,g)} \frac{\binom{N-h}{g-k}\binom{h}{k}}{\binom{N}{g}}.$$

By estimating all the enrichment of the hallmark gene sets for each patient, we obtain a resulting matrix (191 patients × 50 hallmark gene sets). Factor analysis was used to reduce the dimensionality of the resulting matrix. The elements of the matrix were transformed into standardized $z$-scores of $-\log$ ($p$-value) for each hallmark gene set to minimize the different scales between the hallmark gene sets. We determined the range of the optimal factor numbers that satisfy both Kaiser's rule[57] (i.e., all factor numbers should have eigenvalues > 1) and the parallel analysis threshold[58] (i.e., all factor numbers obtained from the parallel analysis should have eigenvalues that are greater than those from the factor analysis) (Supplementary Fig. 5a). We used in this study an optimal factor number, $k = 5$ (see Supplementary Data 1 for other factor numbers). By performing the factor analysis using the optimal values obtained, two matrices were obtained: a loading matrix (Fig. 4a) representing the weight values of the hallmark gene sets for each factor, and a factor score coefficient matrix representing the weights of the identified factors in each patient. Each factor can be characterized with the hallmark gene sets that exceed a typical weight threshold of 0.5 in the loading matrix (Fig. 4a). With the factor score coefficient matrix, we carried out the $k$-means clustering to see which CMS groups are enriched in the clusters (Fig. 4b). To find out the biological

characteristics of each cluster, the statistical significance was assessed for the factor scores of a cluster compared to those of other clusters in each factor.

**Patient classification by the mRMR feature selection method.** By using the mRMR feature selection method[28, 29], several significant hallmark gene sets can be selected to discriminate tumor stages of cancer patients. The mRMR method is an algorithm often used to identify relevant features that are mutually far apart while having a high correlation with the classification variable, for example, the tumor stage in this study. The elements of the matrix resulting from the hallmark gene set enrichment test (191 patients by 50 hallmark gene sets) were transformed into standardized $z$-scores of $-\log(p\text{-value})$ for each hallmark gene set to minimize the different scales between the hallmark gene sets. By applying the mRMR method, 37 hallmark gene sets were selected, which enables the transformed matrix to be reduced to a matrix of 191 patients by 37 hallmark gene sets. From the $k$-means clustering, we found five clusters and examined the statistically significant enrichment of tumor stages in each cluster (hypergeometric test), which resulted in three clusters that have different tumor stage characteristics. To determine the phenotypic characteristics of each cluster, we selected 12 hallmark gene sets that are statistically significant in all three clusters ($-\log(p\text{-value}) > -\log(0.05)$) (Supplementary Fig. 22). From the comparison of the statistical significance of the three clusters in each significant hallmark gene set, we finally found biologically meaningful hallmark gene sets that show significant differences in the statistical test results between a group with the highest $-\log(p\text{-value})$ and the other two groups (Wilcoxon rank-sum test).

**Immune scoring system based on hallmark gene sets.** We investigated whether the immune score based on the hallmark gene set analysis is well distinguishable between MSI and MSS patients. For this, we introduced three immune scores that are defined with the statistical significance of specific hallmark gene sets associated with immune response, such as the immune process category from MSigDB, the immune response factor from our factor analysis, and carefully selected hallmark gene sets that are considered essential for the immune response. The immune scores that we defined show a statistically significant difference between the MSI and MSS patients (Supplementary Fig. 14a). We compared our MSI/MSS classification with the groupings based on several estimates related to immune response or tumor purity including (i) ESTIMATE[59], which represents the fraction of stromal and immune cells in tumor samples, (ii) Immune score[59], which is a basis for the ESTIMATE score and predicts the level of infiltrating immune cells, (iii) Leukocyte score[60], which is strongly associated with microsatellite instability, (iv) ABSOLUTE[61], which estimates tumor purity in cancer samples, (v) the consensus measurement of purity estimations (CPE)[62] incorporating previous tumor purity estimates, and (vi) image analysis of hematoxylin and eosin stain slides (IHC)[62]. While the immune response scores, such as the Immune score, Leukocyte score, and ESTIMATE, were significantly higher in the MSI than in the MSS patients (Supplementary Fig. 14b), the tumor purity estimates, such as ABSOLUTE and CPE, were significantly lower in the MSI than in the MSS patients; however, no significant difference in IHC was observed between the two patient groups (Supplementary Fig. 14c). Therefore, our results indicate that the immune scoring system based on the hallmark gene set, which takes into account both the mutation and expression profiles, can discriminate MSI and MSS colon cancer patients.

**The GPC size according to mutation selection rules.** We applied three different mutation selection rules to identify the order of somatic mutations during tumorigenesis for each patient and then investigated the order of the most commonly observed key driver mutations in colorectal cancers, APC, TP53, KRAS, PIK3CA, and SMAD4. To estimate the order of a pair of driver mutations in each patient, patients who had more than two driver mutations were considered, and patients who had more than 65 somatic mutations were excluded to reduce computational complexity, which finally left 129 patients for the analysis. For individual patients, we performed the following processes to simulate the change in the size of the GPC according to the rules (see Supplementary Fig. 15 for flowchart of the process). Step 1: We selected an initial mutation among all the mutations except driver mutations. Step 2: According to each rule, we determined the next mutation to be added at each evolution time step. Step 3: We repeated the previous steps starting from all the available initial mutations and finally obtained as many mutation sequences as the order of the total number of somatic mutations of the patient. Step 4: By investigating the order of a pair of key driver mutations in the resulting mutation sequences across patients, we constructed a matrix that exhibits the number of mutation sequences such that one driver mutation in a row occurs earlier than the other driver mutation in a column (Fig. 5d–f). Step 5: For each mutation sequence of patients, we calculated the size of the GPC along with the accumulation of somatic mutations (Fig. 5h).

**Description of the mutation selection rules.** The first mutation selection rule was to choose the next mutation that minimizes the overlap with all the previous mutations (Fig. 5a). The degree of overlap between a pair of genes is determined only by their topological properties without applying the network propagation, i.e., the shortest path length between them and their node degrees (Supplementary Fig. 16). The overlap index based on the network propagation, such as the Jaccard

index in Fig. 2c, cannot measure the degree of overlap when two mutation-propagating modules are not overlapped, whereas the overlap index based on the network topology can measure the probability of two mutations to be overlapped even when two mutations are located extremely far from each other. The second rule was to select the next mutated gene such that the size of the connected module between a mutation candidate and all the previously mutated genes was maximized (Fig. 5b). When all mutation candidates did not form any connected modules with previous mutations, the next mutation was randomly selected among the mutation candidates. We used the threshold, $V = 0.001$, to determine the GC. The last rule was to select the next mutated gene that minimized the size of the connected module between a mutation candidate and all the previously mutated genes among mutation candidates overlapped with any previously mutated ones. In the case of the last rule, we first screened mutation candidates in which their mutation-propagating modules overlapped with those of previously mutated genes. Among them, we selected the next mutation such that the size of the connected module between a mutation candidate and all the previously mutated genes were minimized (Fig. 5c). In the case when all mutation candidates did not form any connected modules with previous mutations, the next mutation was randomly selected among the mutation candidates. We used the threshold, $V = 0.001$, to determine the GC.

**Influence of tumor heterogeneity on the formation of a GPC**. We investigated whether the GPC size depends on the tumor heterogeneity by using the mutation profiles obtained from the multiregion biopsies of the primary regions from colorectal cancers with whole-exome sequencing[46]. Among five colorectal cancer patients, we selected two patients, CRC2 and CRC3, both of which have five subclones in each primary tumor. Although they have the same number of subclones, they are different in terms of the tumor heterogeneity because they have different mutation patterns. The proportion of private mutations, a set of mutations that are found in some subclones, of CRC2 is 63% of all the mutations, which is slightly higher than that of CRC3 (60%) (Supplementary Fig. 20a). Moreover, more than half of the cancer genes of CRC2 are contained in the private mutation pool (6 of 11, 55%), whereas in CRC3, only one cancer gene is included in the private mutation pool (1 of 4, 25%). These differential mutation patterns between CRC2 and CRC3 indicate that the CRC2 sample seems to be more heterogeneous than the CRC3 sample. Next, we examined the size of the GPC for individual subclones and that for the bulk, which includes all the mutations of the subclones. Supplementary Fig. 20b shows that the GPC size of the CRC2 bulk increased by 23% compared to the union of the GPCs of the subclones, whereas that of CRC3 increased by 15%. This difference can be understood by considering two extreme examples, two subclones having the exact same mutation profiles and two subclones having mutually exclusive mutation profiles. In the former case (perfect homogeneity), there will be no increase in the GPC size of the bulk because the bulk also has the same mutation profile as the subclones. However, in the latter case (perfect heterogeneity), the GPC size of the bulk can increase significantly if the two GPCs of the subclones are close enough but do not overlap. But both extreme cases are unrealistic. Indeed, subclones share some mutations, but many mutations are often distributed in distinct signaling pathways between different subclones. Then, the GPC of the bulk will be larger than the union of the GPCs of the subclones because although there is no actual interaction between the signaling pathways in different subclones, the signaling pathways appear to cooperate with each other in the bulk case leading to a larger GPC. Therefore, we can conclude that as the tumor heterogeneity of a bulk tumor increases, the effect on the formation of the GPC becomes larger.

**Data availability**. All relevant data and codes are available from the authors.

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

## Acknowledgements

This work was supported by the National Research Foundation of Korea (NRF) grants funded by the Korea Government, the Ministry of Science and ICT (2017R1A2A1A17069642 and 2015M3A9A7067220).

## Author contributions

K.-H.C. designed the project and supervised the research; D.S., J.L., J.-R.G. and K.-H.C. performed the modeling and analysis; and D.S., J.L. and K.-H.C. wrote the manuscript.

## Additional information

**Competing interests:** The authors declare no competing financial interests.

