## [Peer Review File · Nature Communications]

Reviewers' comments:

Reviewer #1 Expert in computational biology:

Many thanks for giving me the opportunity to review this intriguing paper. The authors deployed a complex network biology strategy to dissect the importance and co-occurrence of mutations in colon cancer, and based on their results propose a so-called hypothesis of percolation transition cluster in colon cancer. The idea is intriguing, but I am afraid the study lacks still many elements to judge it suitable for publication.

My major concerns are the following:

1. Patient data selection. Having in mind the massive amount of data available for colon cancer, is it not clear to me why the authors selected the data set used in this study. Further the patient data is poorly characterized. It is not indicated, for example, whether/how many patient samples belong to which tumor stage, whether the samples are taken from patients before or after therapy, and which therapy they underwent. This has to be explained, if they selected the samples for a given subset of patients, it has to be stated. If they use a heterogeneous set of patient data, this has to be justified and the consequences in the analysis have to be discussed.
2. The hypothesis derived is not discussed or compared in detail to state of the art hypothesis/methods based on gene mutations and expression profile to classify colon cancer tumors. See for example: <https://www.ncbi.nlm.nih.gov/pubmed/26457759> or <https://www.ncbi.nlm.nih.gov/pubmed/26982367>
3. Further, the hypothesis is not discussed/compared in detail with other network biology-based hypothesis on the role of somatic mutations in cancer like the work briefly mentioned from the Ideker Lab, or the more recent from the Califano lab (<https://www.ncbi.nlm.nih.gov/pubmed/27322546>)
4. The potential effect of tumor heterogeneity, with the mutations distributed in different clones, is not considered in the context of the analysis, so it is not clear whether the formation of the percolation giant cluster will be affected by this key feature.
5. The hypothesis of the formation of a percolation giant cluster should be validated. It should be validated against a different colon cancer data set, and as I said there is plenty of large datasets available. But one should also investigate whether such a gigantic network structure exists in other types of solid tumors with similar features.
6. Sadly, it is not provided any fraction of the code used for performing the analysis. This essentially preclude any attempt to reproduce by other groups the results here shown, but also would make not straightforward at all for others to use the methodology here described with other data sets or tumor entities.
7. Thinking on the results itself, what drives my attention is the large difference in the size of the percolation cluster from patient to patient (Fig 1c) but also that a large fraction of the patients (78) were not classified in stage I-III. More discussion about this is necessary to make a solid decision.
8. The statements contained in paragraph 357-364 on the use of the hypothesis for therapy design are extremely vague. They should be either written in a clearer style or better removed.

Reviewer #2 Expert in biological networks:

Dear Editor,

In ms NCOMMS-16-26509, the authors present a network-based study of tumorigenesis in colorectal cancer. Their approach captures a progression of many mutations, and shows a percolation effect where the cumulative effect of the mutations transitions suddenly from relatively benign to cancerous. Their work provides insight into the fundamental processes by which cancerous tumors form, and thereby provides insight into potential therapies. It appears that their approach could be mapped directly to other forms of cancer.

This is an interesting and very relevant work. While I could not evaluate the authors' code, I found their method descriptions to be technically correct, and their results are summarized nicely in very informative figures. The text itself was quite readable. Overall, I judge this to be a very strong manuscript.

However, I do have several comments that should be addressed prior to publication. My comments are mostly minor and address the clarity of the manuscript; some notation appears to be used inconsistently and one term (the coverage) would benefit from a more detailed definition. Somewhat more significantly, some aspects of the authors approach are not clear and require some additional explanation (comment 6).

I hope the authors find these comments useful as they revise their work.

Comment 1

Lines 88-89: The authors write, "In this study, we found that the cooperative mutation effects represented by a large connected component in a PPI network form a giant cluster."

This is not as clear as it could be. A cluster is colloquially thought of as a collection of nodes, but in a signaling network the nodes are specific molecules. In what way do the mutation effects form a cluster?

This is clarified later in the ms, of course, but a reader coming to this could be confused here.

Comment 2

On line 157 the authors refer to "mutation pairs". However, if my reading is correct, they are actually referring to pairs of "mutation-propagating modules", as defined on lines 147-148. I reason this because J and C are defined in terms of the modules, and these scores are used to categorize the quantities referenced again on line 157.

If my reading is correct, a consistent notation should be used; otherwise, clarification is needed.

Comment 3

The definition for coverage should be spelled out more clearly. From the diagram in Fig. 2C, it appears as if $S(A,B)$ indicates the set of all elements that exist in A, B, or both. However, this is by definition the union of the two sets. It is unclear that the "padded border" is meant to indicate additional nodes in the connected module that are not in A and B.

The in-text definition on line 623 says that $S(A,B)$ "indicates the size of a connected module between a pair of mutation-propagating modules". This might be clearer if written e.g. "...indicates the size of the connected module in which a pair of mutation-propagating modules A and B exist."

It may be also useful to explicitly address the situation where a connected module has three

mutation-propagating modules, A, B, and C. In this case I infer that $S(A,B)$ would be (A union B union C) and that the coverage would be > 1 . I infer also that the coverage would be 0 if A and B are not members of a connected module, but otherwise the coverage is always > 1 .

If this is correct, I suggest the authors include a sample module C in the Fig and re-label it accordingly. In any case, the definition and surrounding text (in the caption and/or main text) should be expanded to make the definition of coverage more clear.

Comment 4

Lines 396 - 397: The authors write, "Patients with more than 300 mutations were discarded, which left 198 patients." I recommend the authors add a sentence explaining this choice. (perhaps, as indicated in a similar statement in the SI, this is to reduce the computational complexity? If so, do the authors expect any qualitative changes to their results if more are considered?)

Comment 5

In panel 2c the authors indicate that the Fisher's combined test p-value is on the order of e^{-159} . I am not an expert statistician, but to my eye this seems egregiously small and well below the floating point precision of typical computers (which draws into question the validity of the value). While it isn't uncommon for a p-value that is essentially 0 to spit out a very large negative exponent like this, in my view it is normally safer to simply say it is "much much less than" some more reasonable threshold, such as e^{-10} .

That being said, I won't object if the authors are confident that this value is a true representation of the output of the test.

(Note that this comment applies also to a number of figures in the SI.)

Comment 6

What is the value of alpha (line 431)? How was the value chosen? How do the authors' results depend on this choice?

Minor comments

Line 145: "closer" - "closeness"?

Line 214: "Fig. 3e and F" - "Fig. 3e and f"

Line 604: There is a rendering/symbolic error; I see a white square before E_i .

The caption of Fig 4 references " $S_{\text{conn}}(i,j)$ ", which I infer is the same as $S(A,B)$ referenced in Fig 2. The same notation should be used in each case (S vs S_{conn}).

Response to the specific comments of Reviewer 1:

[COMMENT #1] Patient data selection. Having in mind the massive amount of data available for colon cancer, is it not clear to me why the authors selected the data set used in this study. Further the patient data is poorly characterized. It is not indicated, for example, whether/how many patient samples belong to which tumor stage, whether the samples are taken from patients before or after therapy, and which therapy they underwent. This has to be explained, if they selected the samples for a given subset of patients, it has to be stated. If they use a heterogeneous set of patient data, this has to be justified and the consequences in the analysis have to be discussed.

[RESPONSE] In our study, we first mapped the gene expression profiles of individual patients to a large-scale protein-protein interaction (PPI) network to obtain patient-specific PPI networks and then, projected the somatic mutation profile of an individual patient to each network to explore phenotypic features embedded in the giant percolated cluster (GPC). For this approach, there are three requirements for the data set. First, the data set should include both gene expression and somatic mutation profiles of colorectal cancer patients. Second, massive somatic mutations identified by whole-exome sequencing are required for the GPC analysis. One of the important implications of our findings is that although driver mutations are essential for triggering the onset of the GPC, passenger mutations also contribute to the formation of the GPC and eventually to the development of cancer. Therefore, a mutation data set profiled by whole-exome sequencing is more suitable for this study than a data set obtained by targeted-exome sequencing which is often used for identifying specific mutations related to cancer. Third, the data set should include clinical outcomes such as tumor stages. Thus, based on these requirements, we chose from among the many genomic data sets The Cancer Genome Atlas (TCGA) data not only because it is the largest publicly available database for

cancer-genomic studies, but also because it provides both mRNA and whole-exome sequencing data in addition to clinical information on the cancer patients. As a result, we selected those patients having less than 300 mutations from the 223 TCGA colorectal cancer patients to discard outliers having a very high number of mutations which result in a GPC covering the entire network (see also our response to COMMENT #4 of Reviewer 2 for details), and we did not extract any subset of patients having specific clinical characteristics. Indeed, we did not indicate in the manuscript whether patient samples were taken before or after therapy and which therapy patients underwent because such clinical information is not directly related to our study. In fact, such information is somehow eventually represented in the expression profiles of the cancer patients. Following the reviewer's comment, we now include a supplementary table which describes the tumor stages, consensus molecular subtype (CMS), and MSI/MSS information (see also our response to COMMENT #2) of the data set we used in the revised manuscript.

[COMMENT #2] The hypothesis derived is not discussed or compared in detail to state of the art hypothesis/methods based on gene mutations and expression profile to classify colon cancer tumors. See for example: <https://www.ncbi.nlm.nih.gov/pubmed/26457759> or <https://www.ncbi.nlm.nih.gov/pubmed/26982367>

[RESPONSE] An important implication in the formation of a GPC in our study is that the cooperation of multiple somatic mutations can activate multiple cancer-related hallmarks that can cooperatively enhance tumorigenesis. For instance, the co-occurrence of a mutation that persistently activates a proliferative signaling and another mutation that induces invasion and metastasis will promote cancer malignancy. Therefore, it is important to find what kind of multiple hallmarks of cancer are enriched in a GPC and more importantly, to find whether such phenotypic features have a real biological significance. In this respect, as the reviewer suggested, we believe that it is very important and valuable to interpret the phenotypic features contained in the GPC with respect to the classification of tumors such as the consensus molecular subtypes (CMSs)¹ of colorectal

cancer or subgroups of microsatellite-instable (MSI)/microsatellite-stable (MSS) tumors
2.

First, we examined how well the GPC-based patient classification matches the previous CMS classification which is known to be the most robust molecular classification currently available for colorectal cancer¹ and further investigated which key biological features the subtypes have (Fig. R1(a)). By propagating all of the mutations of each

Figure R1. Hallmark gene set analysis. (a) Flowchart of the hallmark gene set analysis for identifying hallmark gene sets enriched in the GPC and for classifying cancer patients based on the selected features using factor analysis. (b) The correlation matrix of hallmark gene sets identifies several sets of correlated hallmark gene sets.

colorectal cancer patient, we obtained a gene list included in the corresponding giant cluster and estimated the enrichment of the hallmark gene sets in each patient through the hypergeometric test. To reduce the dimensionality of the resulting matrix of the hallmark gene set enrichment test (191 patients \times 50 hallmark gene sets), we did factor analysis, which is often used in gene expression data for patient clustering as a robust feature selection method^{3,4}, using standardized z scores of $-\log(\text{p-value})$ for each hallmark gene set because the hallmark gene set variables have different scales. For an optimal predefined factor number ($k = 5$) (Fig. S5a and see Methods for details), the correlation matrix of the hallmark gene sets were clustered into four global factors (Fig. R1(b) and see Table S2 for various factor numbers, $k = 3, 4,$ and 5) each of which can be characterized with several hallmark gene sets for a conventional weight threshold of 0.5: Factor 1 for angiogenesis and the metastasis pathway, Factor 3 for the immune response, Factor 4 for the Myc pathway and uncontrolled proliferation, and Factor 5 for the metabolic pathway (Fig. R2(a)). Interestingly, these factors not only contain important features related to cancer cells but also are very similar to the biological characteristics that distinguish each CMS group¹. Factors 1, 3, 4, and 5 correspond to the characteristics of CMS4 (mesenchymal), CMS1 (MSI immune), CMS2 (canonical), and CMS3 (metabolic), respectively. To investigate whether cancer patients can be classified into the CMS group by these factors, we performed statistical clustering (k-means) analysis on the factor scores of the patients. The result shows that the patient population can be clustered into four clusters, and interestingly, each cluster is strongly correlated with a distinct CMS group (Cluster 1 – CMS2, Cluster 2 – CMS4, Cluster 3 – CMS1, and Cluster 4 – CMS3) (Fig. R2(b)). For a biological understanding of the cluster groups, we then examined which Factors are dominant in each cluster (Fig. R2(c)) and compared the results with the biological signatures of the CMS groups¹. Cluster 1 was mainly characterized as Factor 4 (Myc pathway and uncontrolled proliferation) and Factor 5 (metabolic pathway), while Factor 1 (angiogenesis and metastasis pathway) and Factor 3 (immune response) had a relatively low significance, which is in good accordance with

the biological characteristics of the CMS2 group except for Factor 5. Cluster 2 was mainly

Figure R2. Patient classification by the factor analysis and biological interpretation of the patient clusters. (a) The correlation matrix of the hallmark gene sets was clustered into four global factors with biological characteristics. Each bar indicates the loading strength of a hallmark gene set in each factor. Blue (red) bars represent positive (negative) values, and absolute values were used for the negative values. (b) The four clusters were classified with statistical clustering (k-means) analysis of the factor scores of the patients. The hypergeometric test was performed to examine the statistical significance of the enrichment of individual CMS groups in each cluster. (c) Biological interpretation of the clusters with the identified Factors, each of which corresponds to a biological signature of a CMS group. The distribution of the Factor scores of the patients in one cluster was compared to that of the other remaining clusters. P-values were obtained by performing the Wilcoxon rank-sum test.

characterized as Factor 1 (angiogenesis and metabolic pathway) but not as Factor 4 (Myc pathway and uncontrolled proliferation) and Factor 5 (metabolic pathway). These features are very similar with the biological signatures found in CMS4, which showed EMT-related signatures, but have less statistical significance with the signatures associated with Wnt and Myc targets, cell cycle, or metabolism. Cluster 3 was mainly characterized as Factor 3 (immune response) which is a key biological feature of the CMS1 group. Cluster 4 is mainly characterized as Factor 5 (metabolic pathway), which is in agreement with the enrichment of the metabolism signatures in CMS3. Taken together, our GPC-based hallmark gene set analysis enables colorectal cancer patients to be clustered into informative subtypes for which biological features well match those of the previous CMS groups. An interesting observation of our work is that the hallmark gene set analysis based on the GPC can extract all the features of the CMS groups, such as immune responses, proliferation, metabolism, and metastasis even though factor analysis was used, which is a type of unsupervised learning for feature selection or for data reduction. Moreover, the patient classification with these selected features shows very strong correlations with the CMS groups. There might be a few major reasons why our analysis based on somatic mutation profiles shows a consistent result with the CMS grouping derived from the gene expression profiles. First, we considered somatic mutations which are tumor-specific genetic changes, unlike germline mutations. Therefore, we were able to exclude some external factors that may be unrelated to cancer. Second, by considering the mutation influence on the PPI network, we were able to extract genes that might have influences on their expression levels by somatic mutations. A recent study suggested that mutation patterns correlate with global gene expression levels, and co-occurring driver mutation pairs tend to induce greater degrees of overlap in downstream transcriptional changes⁵. Our result shows that considering cancer-related or driver mutations alone could not capture all the features of the CMS groups (Fig. S8-S13), suggesting that the so-called passenger mutations will also have important roles in determining the functional activity of genes that are related to tumor progression. Third, the biased propagation of mutation influences based on a patient's gene expression profile better reflects the current state of the patient. Fourth, the introduction of an appropriate threshold to mutation influences enables us to determine an optimal gene set that can capture the inherent characteristics of cancer. Taken together, our results imply that a set of essential genes contained in the

GPC obtained from the network propagation of somatic mutations can capture the important cancer-related features in the gene expression.

Second, we investigated whether the immune score based on the hallmark gene set analysis is well distinguishable between MSI and MSS patients. For this, we introduced three immune scores that are defined with the statistical significance of specific hallmark gene sets associated with immune response, such as the immune process category from MSigDB, the immune response factor from our factor analysis, and carefully selected hallmark gene sets that are considered essential for the immune response. The immune scores that we defined show a statistically significant difference between the MSI and MSS patients (Fig. R3(a)). We compared our MSI/MSS classification with the groupings based on several estimates related to immune response or tumor purity including (i) ESTIMATE⁶, which represents the fraction of stromal and immune cells in tumor samples, (ii) Immune score⁶, which is a basis for the ESTIMATE score and predicts the level of infiltrating immune cells, (iii) Leucocyte score⁷, which is strongly associated with microsatellite instability, (iv) ABSOLUTE⁸, which estimates tumor purity in cancer samples, (v) the consensus measurement of purity estimations (CPE)⁹ incorporating

Figure R3. Comparison of immune scores between the MSI and MSS groups. Three types of immune scores were used based on the Hallmark gene set (a), immune signature (b), and tumor purity (c). P-values were obtained with the t-test.

previous tumor purity estimates, and (vi) image analysis of hematoxylin and eosin stain slides (IHC)⁹. While the immune response scores, such as the Immune score, Leukocyte score, and ESTIMATE, were significantly higher in the MSI than in the MSS patients (Fig. R3(b)), the tumor purity estimates, such as ABSOLUTE and CPE, were significantly lower in the MSI than in the MSS patients; however, no significant difference in IHC was observed between the two patient groups (Fig. R3(c)). Therefore, our results indicate that the immune scoring system based on the hallmark gene set, which takes into account both the mutation and expression profiles, can discriminate MSI and MSS colon cancer patients.

We appreciate the valuable comments that have improved our results and added the relevant analysis to the Results and Discussion sections in the revised manuscript.

[COMMENT #3] Further, the hypothesis is not discussed/compared in detail with other network biology-based hypothesis on the role of somatic mutations in cancer like the work briefly mentioned from the Ideker Lab, or the more recent from the Califano lab (<https://www.ncbi.nlm.nih.gov/pubmed/27322546>)

[RESPONSE] Considering the functional consequences of somatic mutations in cancer under the molecular network framework can be a promising strategy for not only understanding the cooperative effects of multiple somatic mutations during tumorigenesis but also for identifying optimal targets for anti-cancer therapies. In a recent study by the Ideker lab¹⁰, Hofree *et al.* have successfully stratified cancer patients based on a network propagation method. However, this network-based analysis differs from ours in many respects, especially in examining the dynamical change of cooperative mutation effects in tumorigenesis. First, we focused on an effective boundary of mutation influences around the mutated nodes and thus, could extract a sub-network for a mutation or a giant component for multiple mutations. Although the influence scores for a mutation can be assigned to all the nodes in the PPI network as in the study by Hofree *et al.*, the mutation could not successfully affect the entire network but could at most cover a few layers of the nearest neighbors. Therefore, by considering such an effective boundary, we were able to obtain a GPC for the mutation profile of each patient and further confirmed the critical transition of the GPC during the development of cancer. Second, in addition to

the mutation profile, the gene expression data of patients were also reflected in the PPI network, thereby obtaining the patient-specific network. Third, by spreading the influence of somatic mutations over the patient-specific network, we were able to extract key phenotypic characteristics that can explain the cancerous state of each patient and to observe phenotypic changes during the cancer development. The VIPER (virtual inference of protein activity by enriched regulon analysis) algorithm from the Califano lab¹¹ is a new regulatory-network based approach for an accurate assessment of protein activity from gene expression data based on ARACNe, an algorithm for reconstructing cellular networks. Although VIPER enables us to evaluate the functional relevance of somatic mutations by inferring aberrant activities induced by mutations, it has limitations in estimating the influence of somatic mutations in the molecular interaction network. Moreover, VIPER does not examine the cooperative process of multiple somatic mutations in tumorigenesis. Therefore, VIPER is totally different from our method in terms of investigating the role of somatic mutations in cancer. Following the reviewer's comment, we included this point in the Discussion section of the revised manuscript.

[COMMENT #4] The potential effect of tumor heterogeneity, with the mutations distributed in different clones, is not considered in the context of the analysis, so it is not clear whether the formation of the percolation giant cluster will be affected by this key feature.

[RESPONSE] The accumulation of mutations in tumorigenesis causes a cancer to diverge into different clones, resulting in subclonal diversification of the tumor cells^{12, 13}. Recent studies in breast cancer¹² and 12 different types of cancers from the pan-cancer analysis¹⁴ showed that the mutation profiles can be very different among the subclones. Multiregion sequencing studies have also shown that differential mutation profiles exist across the subregions in some tumor samples, although some driver mutations are shared in different subclones^{13, 15}. Therefore, we can predict that the formation of a GPC will be affected by the tumor heterogeneity because the formation of a GPC is determined by the mutation profile of a cancer. To explore this, we further investigated whether the GPC size depends on the tumor heterogeneity by using the mutation profiles obtained from the

multiregion biopsies of the primary regions from colorectal cancers with whole-exome sequencing¹⁶. Among five colorectal cancer patients, we selected two patients, CRC2 and CRC3, both of which have five subclones in each primary tumor. Although they have the same number of subclones, they are different in terms of the tumor heterogeneity because they have different mutation patterns. The proportion of private mutations, a set of mutations that are found in some subclones, of CRC2 is 63% of all the mutations, which is slightly higher than that of CRC3 (60%) (Fig. R4(a)). Moreover, more than half of the cancer genes of CRC2 are contained in the private mutation pool (6 of 11, 55%), whereas in CRC3, only one cancer gene is included in the private mutation pool (1 of 4, 25%). These differential mutation patterns between CRC2 and CRC3 indicate that the CRC2 sample seems to be more heterogeneous than the CRC3 sample. Next, we examined the size of the GPC for individual subclones and that for the bulk which includes all the mutations of the subclones. Figure R4(b) shows that the GPC size of the CRC2 bulk increased by 23% compared to the union of the GPCs of the subclones, whereas that of CRC3 increased by 15%. This difference can be understood by considering two extreme examples, two subclones having the exact same mutation profiles and two subclones having mutually exclusive mutation profiles. In the former case (perfect homogeneity), there will be no increase in the GPC size of the bulk because the bulk also has the same mutation profile as the subclones. However, in the latter case (perfect heterogeneity), the GPC size of the bulk can increase significantly if the two GPCs of the subclones are close enough but do not overlap. But both extreme cases are unrealistic. Indeed, subclones share some mutations, but many mutations are often distributed in distinct signaling pathways between different subclones. Then, the GPC of the bulk will be larger than the union of the GPCs of the subclones because although there is no actual interaction between the signaling pathways in different subclones, the signaling pathways appear to cooperate with each other in the bulk case leading to a larger GPC. Therefore, we can conclude that as the tumor heterogeneity of a bulk tumor increases, the effect on the formation of the GPC becomes larger. If whole-exome sequencing of individual subclones is publicly available, we can analyze both the GPC for each subclone and the cancer hallmarks contained within it, which will provide further insight into understanding tumor heterogeneity and cancer evolution. Following the reviewer's comment, we have included this point in the Discussion section of the revised manuscript.

a Number of mutations

CRC2	Number of mutations	Cancer genes
Common	87 (37%)	5 (45%)
Private	144 (63%)	6 (55%)

CRC3	Number of mutations	Cancer genes
Common	32 (40%)	3 (75%)
Private	48 (60%)	1 (25%)

b GPC size

All clones (5178)

GPC(bulk) – Union of GPC(subclones)
Union of GPC(subclones)
23 % increase

All clones (1865)

GPC(bulk) – Union of GPC(subclones)
Union of GPC(subclones)
15 % increase

Figure R4. Influence of tumor heterogeneity on the formation of a GPC. (a) The Venn diagram shows the distribution of the mutation profiles of five subclones in the CRC2 (left) and CRC3 (right) samples. Each number represents the number of mutations that some clones share. (b) The Venn diagram shows the distribution of the gene list contained in the GPC of five subclones in the CRC2 (left) and CRC3 (right) samples. Each number represents the number of genes that some clones share in their GPC. The square box indicates the GPC of the bulk which includes all the mutations of the subclones. The average expression profile of TCGA colorectal cancer patients was used to extract the colon cancer specific average PPI network.

[COMMENT #5] The hypothesis of the formation of a percolation giant cluster should be validated. It should be validated against a different colon cancer data set, and as I said there is plenty of large datasets available. But one should also investigate whether such a gigantic network structure exists in other types of solid tumors with similar features.

[RESPONSE] Following the reviewer's comment, we have further investigated the formation of a GPC for other type of colon cancer (DFCI data set obtained from cBioPortal, $n = 526$)^{17, 18} as well as eight other types of solid tumors (BLCA, BRCA, HNSC, KIRC, LUAD, LUSC, PAAD, and STAD obtained from TCGA). In the case of colorectal cancer, because there exists no publicly available large-scale database, except for the TCGA database which provides both the gene expression and somatic mutation profiles at a genome-wide level, the DFCI data set was used which includes the somatic mutation profiles for a large population of patients but lacks the gene expression information (see also our response to COMMENT #1). To compensate for the lack of gene expression information, the average expression profile of the TCGA colorectal cancer patients was used to extract the colon cancer specific average PPI network. We compared the size of the giant cluster between the colorectal cancer patient and the random expectation for which the same number of virtually mutated nodes as in each patient were randomly generated in a repeated way ($n = 1000$), and the averaged giant cluster size was measured for each patient. Performing this comparison clearly shows that colorectal cancer can be characterized as having a significantly larger size of giant cluster (Fig. R5). We also confirmed that the statistical significance of this tendency was observed in the other types of cancer as well. As a result, we validated our hypothesis of the formation of a GPC in various types of cancers, and we included this further validation result in the Results section of the revised manuscript.

Figure R5. The formation of a GPC in various solid tumors. The data that we considered include colorectal cancer (CRC from the DFCI data set), urothelial bladder carcinoma (BLCA from the TCGA data set), breast invasive carcinoma (BRCA from the TCGA data set), head and neck squamous cell carcinoma (HNSC from the TCGA data set), kidney renal clear cell carcinoma (KIRC from the TCGA data set), lung adenocarcinoma (LUAD from the TCGA data set), lung squamous cell carcinoma (LUSC from the TCGA data set), pancreatic adenocarcinoma (PAAD from the TCGA data set), and stomach adenocarcinoma (STAD from the TCGA data set). The size of the GPC normalized by the number of mutations of each cancer patient was compared to the random expectation for which the same number of mutations for each patient was randomly selected, and the averaged size of the normalized giant clusters ($n = 100$) was used. P-values were obtained with the t-test. The threshold of the mutation influence, $V=0.005$, was used.

[COMMENT #6] Sadly, it is not provided any fraction of the code used for performing the analysis. This essentially preclude any attempt to reproduce by other groups the results here shown, but also would make not straightforward at all for others to use the methodology here described with other data sets or tumor entities.

[RESPONSE] Following the reviewer's comment, we provided all the codes used in our analysis in the Supplementary data of the revised manuscript.

[COMMENT #7] Thinking on the results itself, what drives my attention is the large difference in the size of the percolation cluster from patient to patient (Fig 1c) but also that a large fraction of the patients (78) were not classified in stage I-III. More discussion about this is necessary to make a solid decision.

[RESPONSE] Our results show that as a number of somatic mutations accumulate at various places in the PPI network during the development of cancer, the resulting mutation-propagating modules can form a connected module and finally develop a GPC at a certain critical point. Therefore, to investigate such a critical transition in a single patient during the development of cancer, it might be important to trace the changes in the size of the giant cluster along with the accumulation of somatic mutations. However, when we classify cancer patients or compare their clinical characteristics such as tumor stages, it would be more appropriate to perform the hallmark gene set analysis for the genes that are included in the GPC. In other words, the difference in the size of the GPC between cancer patients is not necessarily related to the patient classification. Figure R6 confirm that there is no significant change in the GPC size depending on the tumor stage. On the other hand, to investigate whether the hallmark gene sets enriched in cancer patients are correlated with clinical tumor stages, we performed statistical clustering analysis of the enrichment score of the hallmark gene sets for cancer patients according to the significant hallmark gene sets. Twelve significant hallmark gene sets were selected

Figure R6. Comparison of the GPC size between groups of different tumor stages. Patients with a GPC size larger than 1500 were excluded from the figure.

to distinguish the tumor stages of cancer patients using the Minimum Redundancy Maximum Relevance (mRMR) feature selection^{19, 20}, an algorithm often used to identify relevant features that are mutually far apart while having a high correlation with the classification variable, for example, the tumor stage in this study. The result revealed that the patient population can be divided into five clusters, and three clusters are strongly correlated with different tumor stages (Cluster 1 - stage 3, Cluster 4 - stage 2, and Cluster 5 - stage 1), although 48 patients in Clusters 2 and 3 do not show any correlation with the tumor stage (Fig. R7(a)). Interestingly, the enrichment of a few hallmark gene sets is significant in each cluster associated with tumor stage, and there are biologically and clinically meaningful relationships between the tumor stages and hallmark gene sets: DNA_REPAIR in stage 1, WNT_BETA_CATENIN_SIGNALING, NOTCH_SIGNALING, and APICAL_JUNCTION in stage 2, and UNFOLDED_PROTEIN_RESPONSE and P53_PATHWAY in stage 3 (Fig. R7(b) and (c)) indicate that there exist tumor stage-specific hallmark gene sets that are enriched in the GPCs of cancer patients. In the early stage of cancer, cancer cells are often initiated by genomic instability due to the dysfunction of DNA repair proteins, and become larger through hyperproliferation. Thus, they start to spread into nearby tissues or lymph nodes when entering the late stages. Our results also show such phenotypic changes according to tumor progression (Fig. R7(d)). A patient cluster characterized as stage 1 tends to show enrichment of the gene set related to DNA repair which is known to cause genomic instability in the early stage of colon cancer^{21, 22}. A patient cluster characterized as stage 2 shows enrichment of the gene set related to the Wnt/ β -catenin and Notch signaling pathways, both of which cooperatively control cell proliferation and tumorigenesis in the intestine²³. A patient cluster characterized as stage 3 shows enrichment of the gene set related to the unfolded protein response, which induces metastasis through hypoxic activation^{24, 25}, and p53 signaling which influences metastasis by the dysfunction of p53²⁶. These results also confirm our hypothesis that the formation of a GPC implies the co-activation of multiple hallmark gene sets that are crucial for tumorigenesis. Taken together, we conclude that cancer patients might be classified into several subtypes according to a few important hallmark gene sets involved in the tumor stages,

consequently suggesting that the GPC conveys biologically and clinically relevant phenotypes. Following the reviewer's comment, we included this analysis in the Results section of the revised manuscript.

Figure R7. Patient classification according to tumor stages and the changes in significant hallmark gene sets depending on the tumor stages. (a) Distribution of the tumor stages in each cluster. Red asterisks indicate statistical significance (hypergeometric test, $p < 0.05$). (b) Comparison of average values of $-\log(p\text{-value})$ of each cluster in individual significant hallmark gene sets. (c) Comparison of the statistical significance ($-\log(p\text{-value})$) of three clusters in each significant hallmark gene set. Red asterisks indicate that there are significant differences in the statistical test results between a group with the highest value and the other two groups (Wilcoxon rank-sum test, $p < 0.05$). Error bars indicate the standard error. (d) Summary of the relationships between the hallmark gene sets and the tumor stages in the individual cluster.

[COMMENT #8] The statements contained in paragraph 357-364 on the use of the hypothesis for therapy design are extremely vague. They should be either written in a clearer style or better removed.

[RESPONSE] By illuminating the GPC, which represents the cooperative effects of the somatic mutations in cancer, our results suggest that identifying optimal targets to fragment the GPC into small pieces can provide a novel therapeutic strategy for the treatment of cancer. The fragmentation of GPC for the cancer treatment not only means breaking the network of the GPC itself, but it also means interfering with cancer-related phenotypes, such as cancer hallmarks, inherent in the GPC. For instance, if a cancer patient has a metastasis-related hallmark enriched in the GPC, we can select minimal targets that can break the GPC among the gene list related to that hallmark. Most cancers have multiple cancer hallmarks that can cooperatively promote tumorigenesis. Thus, we can consider some drug combinations, for example, one that interferes with the uncontrolled proliferation-related hallmark and the other that inhibits the metastasis-related hallmark. Following the comment, we have rephrased the statements on the use of the hypothesis for therapy design in a clearer way in the revised manuscript.

Response to the specific comments of Reviewer 2:

[COMMENT #1]

Lines 88-89: The authors write, "In this study, we found that the cooperative mutation effects represented by a large connected component in a PPI network form a giant cluster."

This is not as clear as it could be. A cluster is colloquially thought of as a collection of nodes, but in a signaling network the nodes are specific molecules. In what way do the mutation effects form a cluster?

This is clarified later in the ms, of course, but a reader coming to this could be liable to be confused here.

[RESPONSE] Following the reviewer's comment, we revised the sentence to clarify its meaning in the revised manuscript as follows: "To explore such a cooperative phenomenon of somatic mutations in tumorigenesis, we employed a network propagation method to measure the spreading of the influence of the somatic mutations on the molecular interaction network and then examined the cooperative effect of the somatic mutations by mapping all of the mutations observed in colorectal cancer from TCGA to a large-scale molecular interaction network. Throughout this network-level systems biological study, we found that a subnetwork area representing the cooperative effect of multiple somatic mutations forms a giant cluster (GC), which is the largest connected subnetwork in which all genes have mutation influence scores beyond a certain threshold."

[COMMENT #2]

On line 157 the authors refer to "mutation pairs". However, if my reading is correct, they are actually referring to pairs of "mutation-propagating modules", as defined on lines 147-148. I reason this because J and C are defined in terms of the modules, and these scores are used to categorize the quantities referenced again on line 157.

If my reading is correct, a consistent notation should be used; otherwise, clarification is

needed.

[RESPONSE] Following the reviewer's comment, we revised this sentence for consistency as follows: "more than half of the mutation-propagation module pairs (55-70%) were synergistic."

[COMMENT #3]

The definition for coverage should be spelled out more clearly. From the diagram in Fig. 2C, it appears as if $S(A,B)$ indicates the set of all elements that exist in A, B, or both. However, this is by definition the union of the two sets. It is unclear that the "padded border" is meant to indicate additional nodes in the connected module that are not in A and B.

The in-text definition on line 623 says that $S(A,B)$ "indicates the size of a connected module between a pair of mutation-propagating modules". This might be clearer if written e.g. "...indicates the size of the connected module in which a pair of mutation-propagating modules A and B exist."

It may be also useful to explicitly address the situation where a connected module has three mutation-propagating modules, A, B, and C. In this case I infer that $S(A,B)$ would be $(A \cup B \cup C)$ and that the coverage would be > 1 . I infer also that the coverage would be 0 if A and B are not members of a connected module, but otherwise the coverage is always > 1 .

If this is correct, I suggest the authors include a sample module C in the Fig and re-label it accordingly. In any case, the definition and surrounding text (in the caption and/or main text) should be expanded to make the definition of coverage more clear.

[RESPONSE] In Fig. 2c, $S(A, B)$ indicates the set of all elements that exist in the connected module when both mutations A and B occur. If A and B are close enough, $S(A, B)$ would be larger than the union of modules A and B. Therefore, as the reviewer pointed out, there can be the "padded border" which indicates additional nodes that do not belong

to any of the modules A and B. But, if A and B are far enough apart such that their modules do not overlap, $S(A, B)$ would be 0. To avoid any confusion, we now more explicitly state the definition of $S(A)$ and $S(A, B)$ as follows: “ $S(A)$ denotes the size of the mutation-propagating module when a mutation A occurs, and $S(A, B)$ indicates the size of the connected module when both mutations A and B occur. If A and B are close enough, $S(A, B)$ would be larger than the union of $S(A)$ and $S(B)$. Therefore, there can be some extended area (gray) which indicates additional nodes that are in neither $S(A)$ nor $S(B)$. However, if A and B are far enough apart such that their modules do not overlap, $S(A, B)$ would be 0”. Moreover, we also replaced “Coverage” with “Synergy” because “C” in Fig. 2c represents the number of additional nodes in connected modules compared to the union of $S(A)$ and $S(B)$. Note that the overlap and synergy are defined only for pairs of two nodes; there is no need to consider a third module in Fig. 2.

[COMMENT #4]

Lines 396 - 397: The authors write, "Patients with more than 300 mutations were discarded, which left 198 patients." I recommend the authors add a sentence explaining this choice. (perhaps, as indicated in a similar statement in the SI, this is to reduce the computational complexity? If so, do the authors expect any qualitative changes to their results if more are considered?

[RESPONSE] From the TCGA colorectal cancer data set, we obtained gene expression data for 263 patients and somatic mutation data for 223 patients. By examining the distribution of the number of mutations, we found that there are a few patients with a very large number of mutations (Fig. R8). Considering that the GPC size of those patients having about 200 mutations already reaches about 80% of the entire network (Fig. S2), it is evident that the GPC of patients having more than 300 mutations will cover the entire network. Hence, such cases will make it difficult to extract any statistically significant hallmark gene set. For this reason, patients with more than 300 mutations were excluded, resulting in 198 patients. Among those, 191 cancer patients for which both mutation and expression profile information are available were finally selected. Following the reviewer’s comment, we fully explained this procedure in the revised manuscript.

Figure R8. Distribution of the number of mutations for 223 cancer patients. The inlet shows the expanded range where the number of mutations is small (red dashed box).

[COMMENT #5]

In panel 2c the authors indicate that the Fisher's combined test p-value is on the order of e^{-159} . I am not an expert statistician, but to my eye this seems egregiously small and well below the floating point precision of typical computers (which draws into question the validity of the value). While it isn't uncommon for a p-value that is essentially 0 to spit out a very large negative exponent like this, in my view it is normally safer to simply say it is "much much less than" some more reasonable threshold, such as e^{-10} .

That being said, I won't object if the authors are confident that this value is a true representation of the output of the test.

(Note that this comment applies also to a number of figures in the SI.)

[RESPONSE] Following the reviewer's comment, we changed the representation of p-values less than e^{-10} as follows: " $p < e^{-10}$ " in both the main and supplementary figures.

[COMMENT #6]

What is the value of alpha (line 431)? How was the value chosen? How do the authors' results depend on this choice?

[RESPONSE] We thank the reviewer for indicating this. α determines the degree of diffusion of a mutation influence throughout the network, and we used an optimal value ($\alpha = 0.7$) for the network constructed from STRING v9.0 which was also used in a previous study by Hofree *et al.*¹⁰. In our study, changing α has a similar effect to changing the threshold V of the mutation influences with respect to the formation of a giant cluster. As α increases for a fixed value of V (or V decreases for a fixed value of α), the size of the resulting giant cluster increases. We considered the cases of various thresholds with a fixed value of α in the analysis of both the giant percolated cluster and the hallmark gene set (Fig. S2, S6, and S7), and confirmed that the main results do not change significantly. Following the reviewer's comment, we have clearly described the value of α in the revised manuscript.

[Minor comments]

Line 145: "closer" - "closeness"?

[RESPONSE] It means "closer". We have clearly rewritten the indicated sentence to avoid any confusion.

Line 214: "Fig. 3e and F" - "Fig. 3e and f"

[RESPONSE] We corrected the typo in the revised manuscript. Thank you for indicating this.

Line 604: There is a rendering/symbolic error; I see a white square before E_i .

[RESPONSE] The indicated error is corrected in the revised manuscript.

The caption of Fig 4 references " $S_{\text{conn}}(i,j)$ ", which I infer is the same as $S(A,B)$ referenced in Fig 2. The same notation should be used in each case (S vs S_{conn}).

[RESPONSE] Following the reviewer's comment, we corrected $S_{\text{conn}}(i,j)$ in the caption of Fig. 4 to $S(i,j)$ for consistency.

References

1. Guinney J, *et al.* The consensus molecular subtypes of colorectal cancer. *Nat Med* **21**, 1350-1356 (2015).
2. Mlecnik B, *et al.* Integrative Analyses of Colorectal Cancer Show Immunoscore Is a Stronger Predictor of Patient Survival Than Microsatellite Instability. *Immunity* **44**, 698-711 (2016).
3. Verhaak RG, *et al.* Integrated genomic analysis identifies clinically relevant subtypes of glioblastoma characterized by abnormalities in PDGFRA, IDH1, EGFR, and NF1. *Cancer Cell* **17**, 98-110 (2010).
4. Iacob E, *et al.* Gene Expression Factor Analysis to Differentiate Pathways Linked to Fibromyalgia, Chronic Fatigue Syndrome, and Depression in a Diverse Patient Sample. *Arthritis Care Res (Hoboken)* **68**, 132-140 (2016).
5. Gerstung M, *et al.* Combining gene mutation with gene expression data improves outcome prediction in myelodysplastic syndromes. *Nat Commun* **6**, 5901 (2015).
6. Yoshihara K, *et al.* Inferring tumour purity and stromal and immune cell admixture from expression data. *Nat Commun* **4**, 2612 (2013).
7. Isella C, *et al.* Stromal contribution to the colorectal cancer transcriptome. *Nat Genet* **47**, 312-319 (2015).
8. Carter SL, *et al.* Absolute quantification of somatic DNA alterations in human cancer. *Nat Biotechnol* **30**, 413-421 (2012).
9. Aran D, Sirota M, Butte AJ. Systematic pan-cancer analysis of tumour purity. *Nat Commun* **6**, 8971 (2015).

10. Hofree M, Shen JP, Carter H, Gross A, Ideker T. Network-based stratification of tumor mutations. *Nat Methods* **10**, 1108-1115 (2013).
11. Alvarez MJ, *et al.* Functional characterization of somatic mutations in cancer using network-based inference of protein activity. *Nat Genet* **48**, 838-847 (2016).
12. Nik-Zainal S, *et al.* The life history of 21 breast cancers. *Cell* **149**, 994-1007 (2012).
13. Sottoriva A, *et al.* A Big Bang model of human colorectal tumor growth. *Nat Genet* **47**, 209-216 (2015).
14. Andor N, *et al.* Pan-cancer analysis of the extent and consequences of intratumor heterogeneity. *Nat Med* **22**, 105-113 (2016).
15. Yates LR, *et al.* Subclonal diversification of primary breast cancer revealed by multiregion sequencing. *Nat Med* **21**, 751-759 (2015).
16. Kim TM, *et al.* Subclonal Genomic Architectures of Primary and Metastatic Colorectal Cancer Based on Intratumoral Genetic Heterogeneity. *Clin Cancer Res* **21**, 4461-4472 (2015).
17. Giannakis M, *et al.* Genomic Correlates of Immune-Cell Infiltrates in Colorectal Carcinoma. *Cell Rep* **17**, 1206 (2016).
18. Gao J, *et al.* Integrative analysis of complex cancer genomics and clinical profiles using the cBioPortal. *Sci Signal* **6**, p11 (2013).
19. Ding C, Peng H. Minimum redundancy feature selection from microarray gene expression data. *J Bioinform Comput Biol* **3**, 185-205 (2005).

20. Radovic M, Ghalwash M, Filipovic N, Obradovic Z. Minimum redundancy maximum relevance feature selection approach for temporal gene expression data. *BMC Bioinformatics* **18**, 9 (2017).
21. Grady WM, Carethers JM. Genomic and epigenetic instability in colorectal cancer pathogenesis. *Gastroenterology* **135**, 1079-1099 (2008).
22. Shih IM, Zhou W, Goodman SN, Lengauer C, Kinzler KW, Vogelstein B. Evidence that genetic instability occurs at an early stage of colorectal tumorigenesis. *Cancer Research* **61**, 818-822 (2001).
23. Fre S, *et al.* Notch and Wnt signals cooperatively control cell proliferation and tumorigenesis in the intestine. *Proc Natl Acad Sci U S A* **106**, 6309-6314 (2009).
24. Mujcic H, *et al.* Hypoxic activation of the unfolded protein response (UPR) induces expression of the metastasis-associated gene LAMP3. *Radiother Oncol* **92**, 450-459 (2009).
25. Wouters BG, Koritzinsky M. Hypoxia signalling through mTOR and the unfolded protein response in cancer. *Nat Rev Cancer* **8**, 851-864 (2008).
26. Powell E, Piwnica-Worms D, Piwnica-Worms H. Contribution of p53 to metastasis. *Cancer Discov* **4**, 405-414 (2014).

REVIEWERS' COMMENTS:

Reviewer #1 (Remarks to the Author):

Many thanks for giving me the opportunity to review again the paper. All in all, the authors have replied with remarkable level of detail to all my comments. I only have two small suggestions to do at this step.

1. Concerning my comment number one, I can conclude that the method proposed requires from the same patients both gene expression and somatic mutation profiles. This seems to be a sufficient motivation to justify the selection of the patients in TCGA made. This could be stated clearly somewhere in the text to avoid any misinterpretation on the procedure to select the data set. Is the choice of the threshold 300 mutations based on any feature of the data set chosen or is it taken from any other previous analyses?

2. The selection of the hallmark gene sets seems to be crucial for this iteration of the manuscript. Could the authors be more precise in terms of the procedure follow to select the genes integrating each hallmark?

The reply given by the authors to the reviewers' comments is the most detailed I have seen to date. I encourage the authors to authorize Nature Communications to provide it as supplementary material.

Response to the specific comments of Reviewer 1:

[COMMENT #1] Concerning my comment number one, I can conclude that the method proposed requires from the same patients both gene expression and somatic mutation profiles. This seems to be a sufficient motivation to justify the selection of the patients in TCGA made. This could be stated clearly somewhere in the text to avoid any misinterpretation on the procedure to select the data set. Is the choice of the threshold 300 mutations based on any feature of the data set chosen or is it taken from any other previous analyses?

[RESPONSE] In response to COMMENT #1 of Reviewer 1 in the previous revision, we have already described the requirements for the dataset as well as the selection procedure of TCGA patients in Supplementary Text. Following the reviewer's comment, we provide further details in the Methods section ("Requirements for datasets") of this revision to avoid any misinterpretation on the procedure. Regarding the choice of the threshold on the number of mutations, we have already described it in the Methods section of the previous version. Following the reviewer's comment, we provide an additional figure in this revision to complement the related content as follows: "By examining the distribution of the number of mutations, we found that there are a few patients with a very large number of mutations (Supplementary Figure 21). Considering that the GPC size of those patients having about 200 mutations already reaches about 80% of the entire network (Supplementary Figure 2), it is evident that the GPC of patients having more than 300 mutations will cover the entire network. Hence, such cases will make it difficult to extract any statistically significant hallmark gene set. For this reason, patients with more than 300 mutations were excluded, resulting in 198 patients. Among those, 191 cancer patients for which both mutation and expression profile information are available were finally selected."

Supplementary Figure 21. Distribution of the number of mutations for 223 cancer patients. The inlet shows the expanded range where the number of mutations is relatively small (red dashed box).

[COMMENT #2] The selection of the hallmark gene sets seems to be crucial for this iteration of the manuscript. Could the authors be more precise in terms of the procedure follow to select the genes integrating each hallmark?

[RESPONSE] Following the reviewer’s comment, we fully explain the procedure of determining the significance of the hallmark gene sets in the Method section of this revision as follows: “To explore the biological functions that are enriched in the giant cluster of each patient, we examined the enrichment of hallmark gene sets by hypergeometric test. For a gene list included in the giant cluster, let h be the number of genes annotated to a certain hallmark gene set, and let N and g be the network size and the number of genes in the giant cluster, respectively. Suppose that the giant cluster has x genes annotated to this hallmark gene set, we can model x by a hypergeometric distribution under the null hypothesis that a gene annotated to the hallmark gene set and

a gene in the giant cluster are independent events. Then, the p-value that measures the significance of enrichment is the probability of observing x or more genes annotated to the hallmark gene set in the giant cluster,

$$\text{p-value} = \sum_{k=x}^{\min(h,g)} \frac{\binom{N-h}{g-k} \binom{h}{k}}{\binom{N}{g}}.$$

By estimating all the enrichment of the hallmark gene sets for each patient, we obtain a resulting matrix (191 patients \times 50 hallmark gene sets).”